# A stress-induced paralog of *Lhcb4* controls the photosystem II functional architecture in *Arabidopsis thaliana*

Roberto Caferri [1,6], Qian Zhou[2,3,6], Luca Dall'Osto [1], Antonello Amelii [1], Jianyu Shan[2,3], Zhenfeng Liu [2,3] & Roberto Bassi [1,4,5] ✉

Photosystem II (PSII) is the pigment-protein complex catalysing light-induced water oxidation. In *Arabidopsis thaliana,* it includes three Lhcb4–6 proteins linking the core complex to peripheral trimeric antennae. While Lhcb5 and Lhcb6 are encoded by single genes, Lhcb4 is encoded by three isoforms: *Lhcb4.1* and *Lhcb4.2*, constitutively expressed, and *Lhcb4.3* (*Lhcb8*), which accumulates under prolonged abiotic stress. Lhcb8 substitutes for Lhcb4, preventing Lhcb6 accumulation and resulting in a smaller PSII with high quantum yield. Cryo-electron microscopy reveals that Lhcb8 has a shorter carboxy-terminal domain, lacks two chlorophylls, and interacts more tightly with the PSII core, inducing structural changes in the PSII antenna system, ultimately inhibiting the formation of PSII arrays and favouring plastoquinone diffusion. We suggest that dynamic Lhcb4 vs Lhcb8 expression allows for PSII acclimation to contrasting light conditions, offering the potential for engineering crops with improved light use efficiency.

Photosynthesis uses sunlight as the primary energy source to reduce $CO_2$ to organic compounds. Eukaryotic phototrophs collect photons through antenna complexes, which consist of a protein backbone equipped with photo-excitable pigments such as chlorophylls (Chl) and carotenoids (Car). The light energy absorbed by these pigments is subsequently funnelled to the reaction centre (RC), where primary photochemical reactions occur.

Light-harvesting and photochemistry are catalysed by photosystems (PSII and PSI), which operate in series. PSII possesses a large antenna system that dynamically adjusts through short and long-term acclimation mechanisms to optimize the conversion efficiency of sunlight into chemical energy[1,2]. From the initial light absorption events to the carbon fixation reactions, multiple regulatory mechanisms ensure operational resilience and prevent damage to the photosynthetic apparatus[3,4]. Indeed, land plants constantly face challenges posed by environmental fluctuations, especially in light quality and intensity, temperature, and water availability. As a result, evolution led

to the expansion of the *Light Harvesting Complex* (*LHC*) multigene family, the sub-functionalization of LHC isoforms[5] and their differential transcriptional activation[6–8].

The higher plant PSII is a dimeric core complex ($C_2$) surrounded by an outer antenna system encoded by *Lhcb1–6* genes. Lhcb4–6 proteins are monomeric, stoichiometric with RC and bridge core complexes to the outer LHCII antenna composed of trimeric assemblies of Lhcb1, Lhcb2, and Lhcb3 subunits[9]. LHCII trimers bind to the PSII core complex with varying affinities and are classified as strongly- (S), moderately- (M), and loosely- (L) associated LHCIIs[10]. While the pigment-protein composition of $C_2$ is highly conserved across species[11], the antenna complexes add to the variability in PSII structure among different organisms[12]. Each LHCII trimer has a specific Lhcb composition[13], influencing its functional properties[14]. Different compositions of PSII SC were shown to directly affect the dynamics of excitation energy transfer[15] and control PSII absorption cross-section in the changing light environment. In *Arabidopsis thaliana*, among

[1]Laboratory of Photosynthesis and Bioenergy, Department of Biotechnology, University of Verona, Verona, Italy. [2]State Key Laboratory of Biomacromolecules, Institute of Biophysics, Chinese Academy of Sciences, Beijing, China. [3]College of Life Sciences, University of Chinese Academy of Sciences, Beijing, China. [4]Accademia Nazionale dei Lincei, Rome, Italy. [5]Stazione Zoologica Anton Dohrn, Villa Comunale, Naples, Italy. [6]These authors contributed equally: Roberto Caferri, Qian Zhou. ✉e-mail: roberto.bassi@univr.it

monomeric Lhcbs, single-copy genes encode Lhcb5 and Lhcb6 poly-peptides, while the Lhcb4 protein is encoded by two constitutively expressed paralog genes, *Lhcb4.1* and *Lhcb4.2*. In addition, a Lhcb4-like protein (hereafter termed Lhcb8) is encoded by the *Lhcb4.3* gene and is only expressed under strong and persistent abiotic stress[16,17]. Upregulation of *Lhcb8* was also observed in *A. thaliana* grown under field conditions vs growth chambers[18]. This suggests that Lhcb8 might play a role in helping plants coping with excess light[19]. Even under high light intensity, Lhcb4 is only partially replaced by Lhcb8, making it challenging to study the physiological role of Lhcb8.

In this study, we generated *A. thaliana* genotypes lacking the *Lhcb4.1*, *Lhcb4.2,* and *Lhcb4.3* genes and complemented them with either *Lhcb4.1* or *Lhcb4.3* sequences, resulting in plants that exclusively accumulate either the Lhcb4.1 or Lhcb8 polypeptide. We report on the structural and functional modifications of PSII that occur depending on which Lhcb4 isoform is incorporated into the PSII supercomplex.

## Results

### The *Lhcb8-only* plants display an altered antenna protein composition

The *koLhcb4.1 koLhcb4.2 koLhcb4.3* triple knock-out genotype (hereafter *koLhcb4*), which lacks the Lhcb4 protein, was complemented with the endogenous *Lhcb4.3* (*Lhcb8*) sequence under the control of the *A.t.Lhcb4.1* promoter[20]. The resulting genotype was referred to as *Lhcb8-only* (Fig. 1a–d). The level of Lhcb8 accumulation was quantified in 30 independent antibiotic-resistant transgenic lines using an anti-Lhcb4 primary antibody, detecting both Lhcb4 and Lhcb8 protein, albeit with different affinity (Supplementary Fig. S1). The range of Lhcb8 accumulation was wide, from very low to a level corresponding to the sum of Lhcb4.1 and Lhcb4.2 polypeptides in the *wild type* (*WT*) (Supplementary Fig. S2), suggesting a stoichiometry 1:1 with PSII core complex[21] (Fig. 1d). The two highest-expressing lines (*Lhcb8-only #1* and *Lhcb8-only #2*) were selected for further experiments, together with two high-expression lines obtained by complementation with an analogous construct expressing Lhcb4.1, which accumulated the protein in *WT* amount (Supplementary Fig. S3)[22].

After four weeks of growth under control conditions (120 μmol photons m$^{-2}$ s$^{-1}$, 23 °C, 8/16 h day/night), the leaf area of the *koLhcb4* genotype was smaller than the *WT*, consistent with previous reports[20] (Fig. 1a). *Lhcb8-only* plants, instead, did not differ from *WT* and *Lhcb4.1-only* (Fig. 1a). Coomassie-stained SDS-PAGE gels (Fig. 1b) showed that in *Lhcb8-only* lines, the Lhcb8 protein migrated at an apparent molecular weight lower than those of Lhcb4.1/4.2 isoforms, as expected for the deduced sequence, shorter by 14 amino acids at the C-terminus. Moreover, the *koLhcb4* plants also lacked Lhcb6, which is restored in the *Lhcb4.1-only* genotype but not in the *Lhcb8-only* lines (Fig. 1c, f).

In the *koLhcb4* genotype, the accumulation of the Lhcb3 subunit of LHCII was reduced by 36% compared to the *WT*. While the *Lhcb4.1-only* line fully restored Lhcb3 levels, the expression of Lhcb3 in the *Lhcb8-only* lines was only partially restored, averaging a 27% ± 5% reduction in the two lines. (Fig. 1c, e and Supplementary Fig. S3). Finally, the accumulation of the major light-harvesting proteins Lhcb1 and Lhcb2 was not significantly different across all genotypes, except for the *koLhcb4* line, which showed over-accumulation of LHCII trimers compared to the *WT*, according to a previous report[20] (Supplementary Fig. S4).

### The PSII supercomplex of *Lhcb8-only* plants differs in molecular size and absorption cross-section with respect to *WT*

The overall organization of the Chl-binding complexes was then investigated via non-denaturing Deriphat-PAGE electrophoresis of thylakoid membranes solubilized with the non-ionic detergent α-dodecyl maltoside (α-DDM, 0.5–1% w/v) (Fig. 2a and Supplementary Fig. S5). The *WT* and the *Lhcb4.1-only* plants both showed four high molecular weight (HMW) green bands corresponding to the $C_2S_2M_2$,

$C_2S_2M$, $C_2S_2/C_2SM$, $C_2S/C_2M$ PSII SC configurations[23]. Notably, the *Lhcb8-only* plants lacked the $C_2S_2M_2$ and $C_2S_2M$ PSII SC but retained the smaller $C_2S$ and $C_2S_2$ PSII SC. The *koLhcb4* genotype, instead, showed only the $C_2$ band, containing the dimeric PSII core complex (Fig. 2a). The LHCII(M)-Lhcb4-Lhcb6 complex was also missing in the *koLhcb4* genotype[24]; it was fully restored in the *Lhcb4.1-only* genotype but not in the *Lhcb8-only* sample. The fluorescence induction kinetics measured in the presence of DCMU showed a slower rise in *Lhcb8-only* lines compared to the *WT* or *Lhcb4.1-only* (Fig. 2b), implying a reduced PSII functional antenna size in *Lhcb8-only* lines as well. Meanwhile, no significant variations were detected in terms of Chl content per leaf area and the Chl *a/b* and Chl (*a* + *b*)/Car ratio (Table 1).

### The PSII supercomplex from *Lhcb8-only* has an expanded and loose architecture

To further identify the differences between the PSII-SCs from the *Lhcb8-only* plants vs *Lhcb4.1-only*, we solved the structures of $C_2S_2$-type PSII SC (Lhcb8-$C_2S_2$ and Lhcb4.1-$C_2S_2$) from the *Lhcb8-* and *Lhcb4.1-only* genotypes through the single-particle cryo-EM method (Fig. 3a, b). Thylakoid membranes were solubilized with 0.8% α-DDM and fractionated by sucrose gradient ultracentrifugation. The HMW green band was harvested and analysed as described in the Supplementary Material and Methods. The overall cryo-EM map resolutions of Lhcb8-$C_2S_2$ and Lhcb4.1-$C_2S_2$ are 3.0 and 3.1 Å, respectively. The cryo-EM data collection, processing scheme, representative cryo-EM densities, and statistics are summarized in Supplementary Figs. S6–S8 and Supplementary Table 3, respectively. Besides the $C_2S_2$-type complex, the solubilized thylakoid membranes of the *Lhcb4.1-only* plants also contained the $C_2SM$, $C_2S_2M$, and $C_2S_2M_2$-type PSII-SCs as shown in Supplementary Fig. S9. In comparison, no $C_2SM$, $C_2S_2M$, and $C_2S_2M_2$-type PSII-SCs (only $C_2S$ and $C_2S_2$) were observed in the sample prepared from the *Lhcb8-only* plants (Supplementary Fig. S6). The characteristic cryo-EM density features for distinguishing Lhcb8 and Lhcb4.1 and those of Lhcb2 and Lhcb1 in S-LHCII are reported in Supplementary Figs. S10 and S11. In both Lhcb8-$C_2S_2$ and Lhcb4.1-$C_2S_2$, the S-LHCII trimers were identified as an (Lhcb1)$_2$Lhcb2 heterotrimer; this is at variance with the case of Lhcb8-$C_2S_2$ from *Picea abies*, which was reported to contain a Lhcb1 homotrimer at the S-LHCII site[25]. In *A. thaliana* Lhcb8-$C_2S_2$ (and Lhcb4.1-$C_2S_2$), the LHCII monomer facing CP43 and in contact with Lhcb8/Lhcb4.1 was assigned as Lhcb2 instead of Lhcb1, whereas the other two in contact with Lhcb5 and at the peripheral position both contained Lhcb1 (Fig. 3a).

Figure 3b showed the superimposed structures of Lhcb8-$C_2S_2$ and Lhcb4.1-$C_2S_2$ using the central D1 subunits as a reference. Remarkably, rearrangements occurred in the antenna domains due to the replacement of Lhcb4.1 by Lhcb8. The Lhcb8-$C_2S_2$ had CP47 and CP43 shifted slightly (by 0.4–1.5 Å) toward Lhcb8 and Lhcb5/S-LHCII, respectively, as compared to the corresponding ones in Lhcb4.1-$C_2S_2$. Moreover, Lhcb8, Lhcb5, and S-LHCII all moved further away from the PSII core, leading to an outward expansion of the Lhcb8-$C_2S_2$ relative to the Lhcb4.1-$C_2S_2$ (Fig. 3b). Previous work showed that a long hairpin motif in the amino-proximal region of Lhcb4 binds to a stromal surface groove on CP47-PsbH and is crucial for the assembly of Lhcb4 with CP47[26]. While Lhcb8 and Lhcb4.1 both contain the long hairpin motif, the one in Lhcb8 formed stronger interactions with CP47/PsbH at two major sites (Tyr93 and Phe95) as compared to the corresponding ones in Lhcb4.1 (Phe92 and Ile94) (Fig. 3c, d). Such enhanced interactions may result in a 3.5° clockwise rotation of Lhcb8 around CP47 compared to Lhcb4.1. This Lhcb8 rotation pushed the S-LHCII' and Lhcb5' on the same side of Lhcb8 to a more peripheral position than the corresponding ones in Lhcb4.1-$C_2S_2$ (Fig. 3b). At the interface between Lhcb8 and CP47, the Chl a603$_{Lhcb8}$-Chl a610$_{CP47}$ and Chl a609$_{Lhcb8}$-Chl a610$_{CP47}$ distances were increased by 0.9 and 0.7 Å respectively as compared to the corresponding pairs in Lhcb4.1-$C_2S_2$. In contrast, the

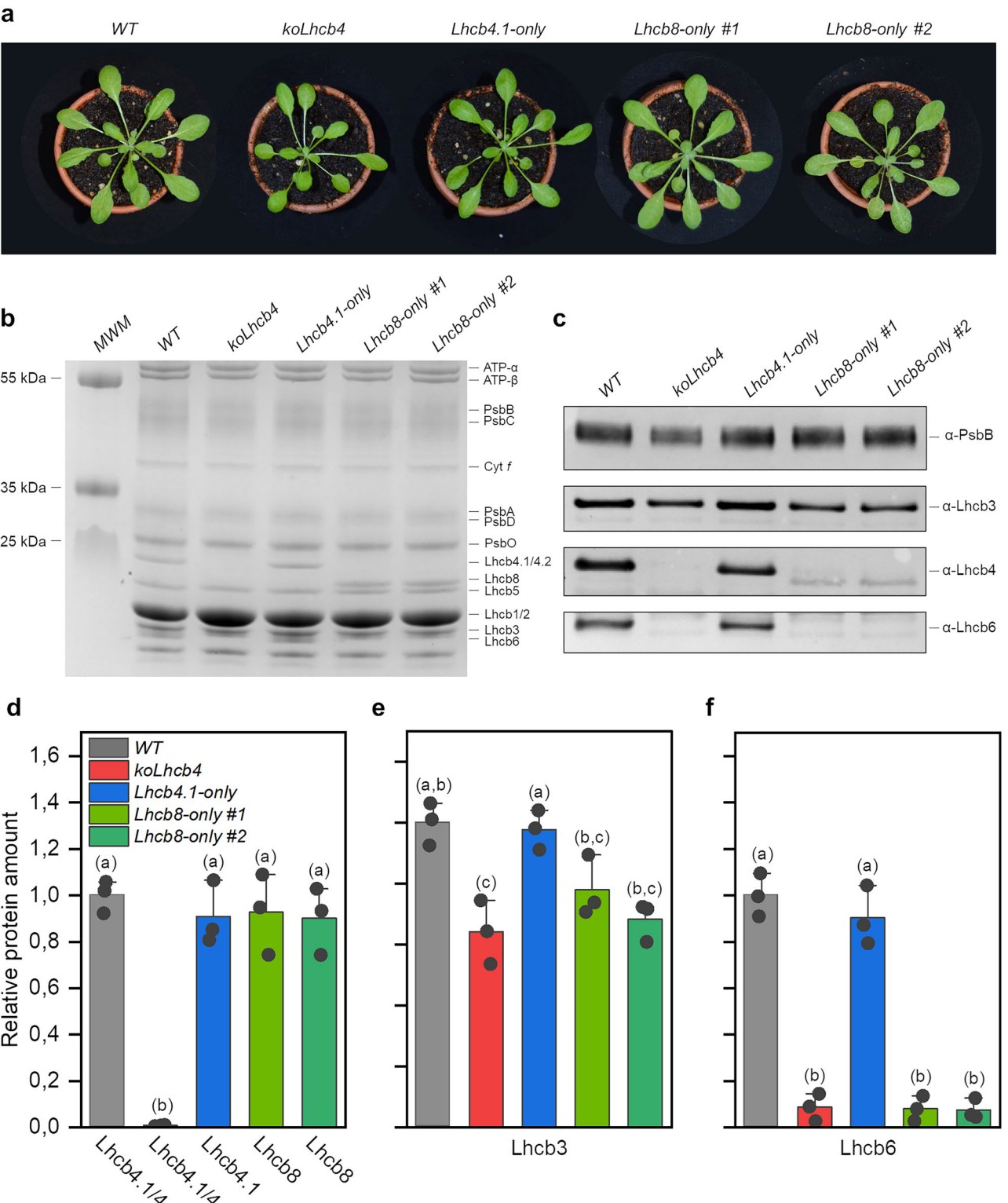

**Fig. 1 | Characterization of *WT*, *koLhcb4*, and *Lhcb8-*/*Lhcb4.1-only* genotypes.** **a** Growth phenotype of *A. thaliana wild type* and mutant plants (*koLhcb4*, *Lhcb4.1-only*, *Lhcb8-only*) in control conditions (120 µmol photons m$^{-2}$ s$^{-1}$, 23 °C, 8/16 h day/night, and 70% humidity) for six weeks. **b** Electrophoretic separation of thylakoid proteins of selected genotypes and Coomassie staining. MWM, molecular weight marker. **c** Immunodecoration of a nitrocellulose membrane developed using anti-PsbB (homemade), anti-Lhcb3 (AS01002, Agrisera), anti-Lhcb4, and anti-Lhcb6 (homemade) antibodies. Equal amounts of Chl (0.5 or 2 µg) of extracted thylakoid proteins were loaded. **d**–**f** Densitometric analysis of Lhcb4 (**d**), Lhcb3 (**e**), and Lhcb6 (**f**) protein levels based on immunodecoration experiments shown in Supplementary Fig. S3 and normalized to the core antenna protein PsbB. Data are shown as mean ± standard deviation of 3 technical replicates. Statistical significance was determined using a one-way ANOVA test followed by the Tukey's test shown with lower-case letters (*P*-value ≤ 0.05).

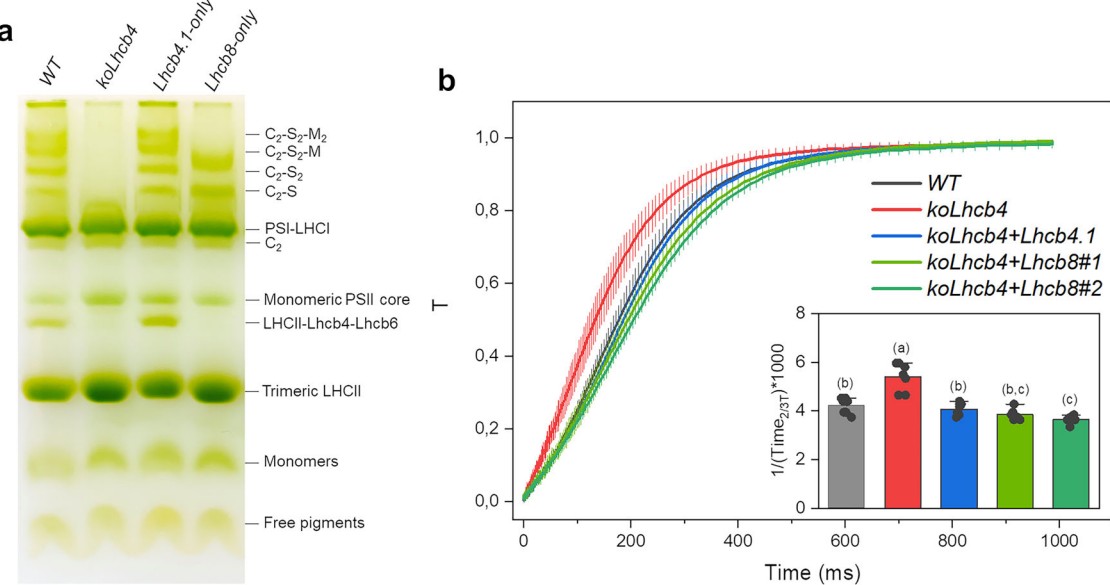

**Fig. 2 | Biochemical and fluorimetric analysis of the *Lhcb8-only* and related genotypes. a** A non-denaturing Deriphat-PAGE (4%–12% acrylamide) was performed to assess the overall organization of Chl-binding proteins. Solubilisation of thylakoid membranes was performed with 0.8% α-DDM (w/v). **b** Fluorescence induction kinetics were recorded in intact leaves that were dark-adapted for 30 min and infiltrated with a buffer containing 50 µM of DCMU (see "Methods" for the detailed procedure). Data are shown as mean ± standard deviation of $n \geq 7$ biological replicates. The statistical significance was assessed by a one-way ANOVA test followed by the Tukey's test (in lower-case) (*P*-value ≤ 0.05).

## Table 1 | Pigment content and fluorescence parameters

| Genotype | Chl *a*/Chl *b* | Chl/Car | µmol Chl (*a* + *b*) / cm² | Fresh weight (g) | $F_O$ | $F_{max}$ | $F_v/F_m$ |
|---|---|---|---|---|---|---|---|
| *WT* | 2.88 ± 0.24 (a) | 4.18 ± 0.12 (a) | 0.14 ± 0.01 (a) | 0.43 ± 0.04 (a) | 0.42 ± 0.02 (a) | 22.57 ± 1.35 (a) | 0.81 ± 0.00 (a) |
| *koLhcb4* | 2.77 ± 0.07 (a) | 4.19 ± 0.20 (a) | 0.12 ± 0.00 (a) | 0.31 ± 0.06 (b) | 0.58 ± 0.02 (b) | 24.51 ± 0.37 (a) | 0.76 ± 0.00 (b) |
| *Lhcb4.1-only* | 2.93 ± 0.07 (a) | 4.32 ± 0.07 (a) | 0.14 ± 0.01 (a) | 0.35 ± 0.07 (a) | 0.43 ± 0.04 (a) | 22.93 ± 2.09 (a) | 0.81 ± 0.00 (a) |
| *Lhcb8-only#1* | 2.93 ± 0.02 (a) | 4.27 ± 0.17 (a) | 0.13 ± 0.00 (a) | 0.35 ± 0.04 (a) | 0.42 ± 0.03 (a) | 21.53 ± 1.71 (a) | 0.80 ± 0.01 (a) |
| *Lhcb8-only#2* | 3.06 ± 0.11 (a) | 4.24 ± 0.19 (a) | 0.15 ± 0.00 (a) | 0.39 ± 0.06 (a) | 0.45 ± 0.02 (a) | 23.26 ± 0.86 (a) | 0.80 ± 0.01 (a) |

Parameters were measured in homozygous T3 plants, 5–6 weeks after germination. Chl and Car amounts were quantified using HPLC analysis (see "Methods" section). Fresh weight refers to six-week-old plants grown under control conditions. $F_O$, $F_{max}$, and $F_v/F_m$ were acquired upon 30 min of dark adaptation. The statistical significance was assessed by a one-way ANOVA test followed by the Tukey's test, shown in lower-case letters (*P*-value ≤ 0.05, $n$ = 3 biological replicates for the pigment estimation and $n \geq 5$ for the fluorescence parameters).

distance between Chl a616$_{Lhcb8/Lhcb4.1}$-Chl a616$_{CP47}$ was unaffected (Supplementary Fig. S12a, b). Similarly, the distance between Chl a614$_{Lhcb2}$ and Chl a506$_{CP43}$ was slightly increased in Lhcb8-C$_2$S$_2$ (Supplementary Fig. S12c, d). The changes in Chl-Chl connections were most dramatic at the Lhcb5-CP43 interface in Lhcb8-C$_2$S$_2$, as the Chl b601$_{Lhcb5}$-Chl a513$_{CP43}$, Chl a602$_{Lhcb5}$-Chl a512$_{CP43}$, and Chl a614$_{Lhcb5}$-Chl a503/Chl a501$_{CP43}$ distances were all increased by 1.5–2.1 Å relative to those in Lhcb4.1-C$_2$S$_2$ (Supplementary Fig. S12e, f). Within the core complexes, the connections between CP47/CP43 and the D1/D2/cytb559 components were not much affected in Lhcb8-C$_2$S$_2$ with respect to Lhcb4.1-C$_2$S$_2$ (Supplementary Fig. S12g–j). As a result of these conformational changes, the efficiencies of energy transfer from Lhcb8, S-LHCII, and Lhcb5 to PSII core might be lower in Lhcb8-C$_2$S$_2$ than those of Lhcb4.1-C$_2$S$_2$.

### PSII efficiency and NPQ response of the *Lhcb8-only* plants

To verify whether the changes detected in the PSII pigment-protein complexes of the *Lhcb8-only* genotype did affect PSII function, we measured PSII quantum efficiency[20]. The $F_v/F_m$ ratio was similar in *Lhcb8-only* and *Lhcb4.1-only* plants and reduced in *koLhcb4*. The distribution of Lhcb8 accumulation levels in the segregating T2 population displayed a quadratic correlation with the $F_v/F_m$ values (Fig. 4b).

This relationship followed a polynomial distribution, where near-zero Lhcb8 levels (~ 0.01) corresponded to a significantly reduced $F_v/F_m$ value (~0.77). Recovery to *WT* $F_v/F_m$ values (0.81) was observed at ~0.9 or higher protein accumulation levels.

Previous work has shown that the loss of Lhcb6, a major feature of *Lhcb8-only* plants, could be reproduced by a single point mutation, H242L, on the Chl b614 ligand of Lhcb4.1[27]. The *koLhcb4* + *Lhcb4.1$_{H242L}$* mutant, however, did not recover to the *wild type* $F_v/F_m$ values, at difference with *Lhcb8-only*. Also, the *koLhcb4* + *Lhcb4.1$_{H242L}$* mutant showed its PSII particles being arranged in a semicrystalline array. This condition has been shown to restrict plastoquinone diffusion, leading to decreased photosynthetic electron transport[20,27].

Upon electron microscopy (EM) inspection of the negatively stained grana membranes prepared from thylakoids, unexpectedly, the *Lhcb8-only* genotype revealed a PSII SC macro-organization with randomly distributed stain-excluding particles in a negatively stained dark background, similar to the *WT* and *Lhcb4.1-only* plants (Fig. 5a, d, f, g) despite the lack of Lhcb6. For *koLhcb4* plants, the PSII particles are also randomly distributed but mostly smaller than those of Lhcb8-only or WT plants (Fig. 5b). A previous work with *koLhcb6* mutant[28], indeed, has shown an organization of PSII into regular arrays, a feature that was not observed in *Lhcb8-only*. We also analyzed the samples from the

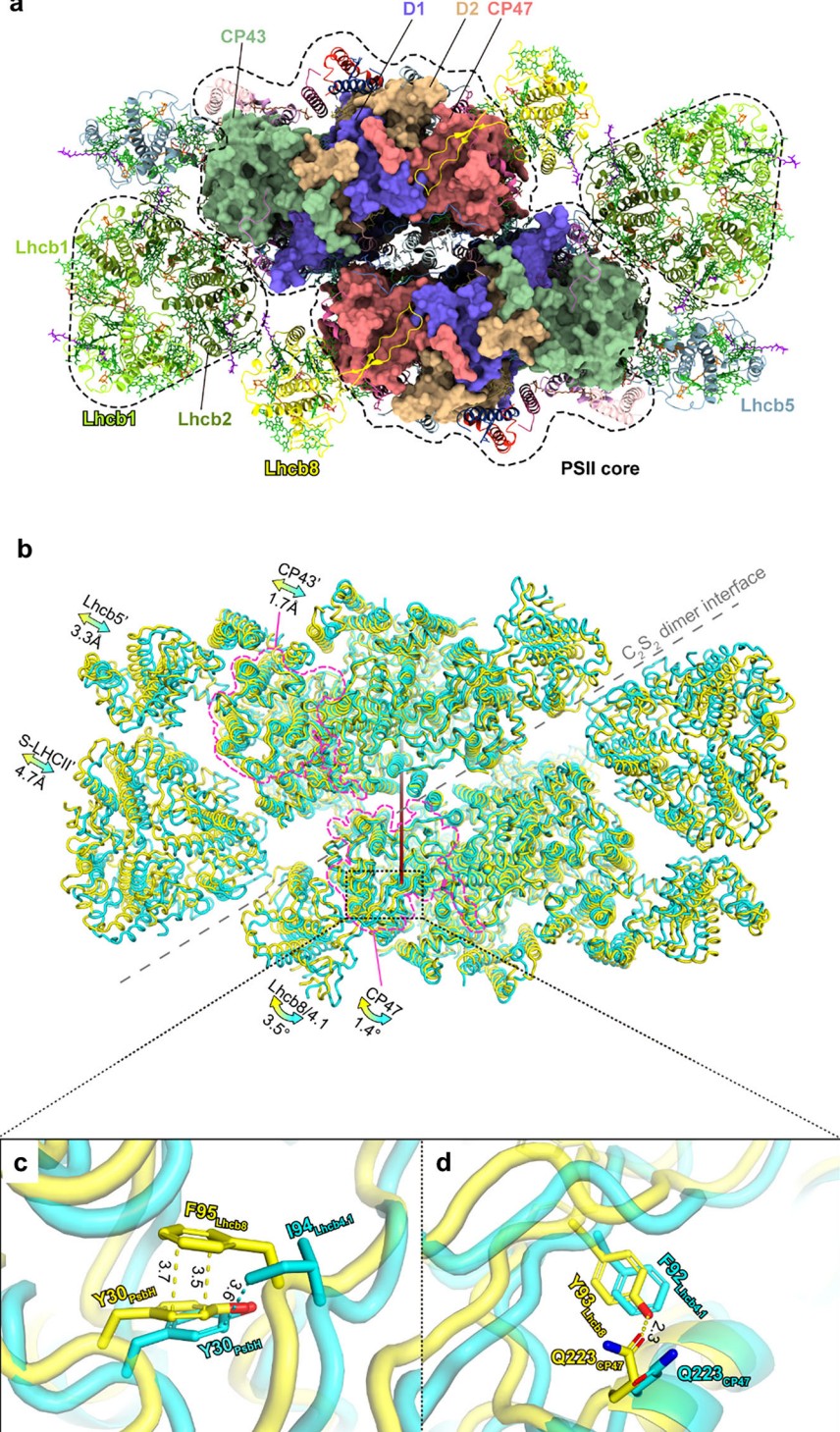

**Fig. 3 | The overall architecture of the Lhcb8-C₂S₂ supercomplex in comparison with the Lhcb4.1-C₂S₂ SC. a** Cryo-EM structure of the Lhcb8-C₂S₂ supercomplex. The four major PSII core subunits are presented as surface models, and the remaining ones are shown as cartoon models. The pigment molecules are shown as stick models. **b** Superposition of Lhcb8-C₂S₂ with Lhcb4.1-C₂S₂. The pigment and lipid molecules were omitted for clarity. The protein subunits in Lhcb8-C₂S₂ and Lhcb4.1-C₂S₂ are all coloured yellow and cyan, respectively. The approximate boundaries of CP47 and CP43 are outlined by the magenta dash lines. The dark red line near the centre of the PSII core dimer represents the axis around which Lhcb8/Lhcb4.1 on the left side was rotated relative to each other. **c, d** Zoom-in views of the two local sites at the amino-proximal regions of Lhcb8 and Lhcb4.1 exhibited clear differences. All the distances labelled near the dash lines are in angstroms (Å).

*koLhcb6* mutant and the *koLhcb4 + Lhcb4.1*$_{H242L}$, both lacking the monomeric antenna Lhcb6 (Fig. 5c, e). The grana membranes displayed highly ordered arrays of PSII particles with a C₂S₂ head-to-tail organisation. This observation implies that the presence of Lhcb8, rather than Lhcb4.1 in PSII supercomplexes, may prevent array

formation (Fig. 5f, g), which is otherwise triggered by the absence of Lhcb6.

Since *Lhcb8* was shown to be expressed under persistent light stress[16–18], we assessed the impact of constitutive Lhcb8 expression on the plant photoprotective capacity. Intriguingly, the *Lhcb8-only* lines

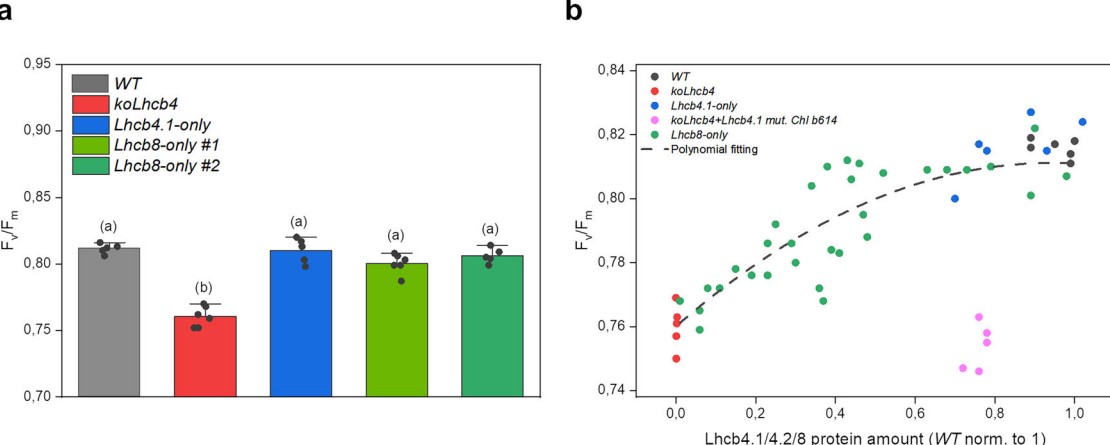

**Fig. 4 | PSII quantum efficiency and its correlation with Lhcb8 levels. a** $F_v/F_m$ values measured using the Dual PAM 100 (Walz) on intact leaves, adapted to the dark for 30 min. **b** Correlation of the $F_v/F_m$ parameter and protein accumulation. The estimation of Lhcb4.1/4.2 or Lhcb8 accumulation was based on densitometric analysis of western blot signals on nitrocellulose membranes using a homemade anti-Lhcb4 antibody. The Lhcb8 quantification value was adjusted based on the cross-reactivity test described in Supplementary Fig. S1. Data are shown as mean ± standard deviation of $n \geq 5$ biological replicates. The statistical significance was assessed by a one-way ANOVA test followed by the Tukey's test and shown on lower-case letters ($P$-value ≤ 0.05).

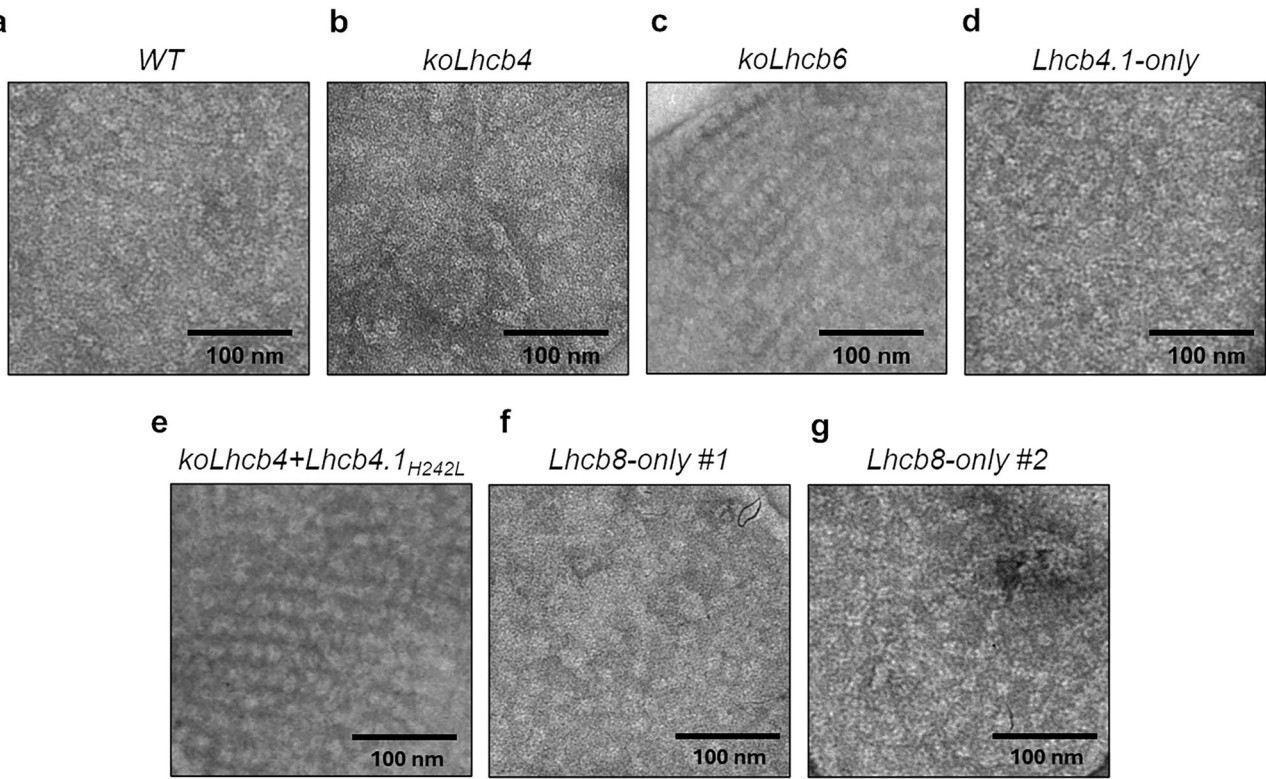

**Fig. 5 | Micrographs of grana partitions from isolated thylakoids. a** *WT*; **b** *koLhcb4*; **c** *koLhcb6*; **d** *Lhcb4.1-only*; **e** *koLhcb4 + Lhcb4.1$_{H242L}$*; **f** *Lhcb8-only #1*; **g** *Lhcb8-only #2*. Thylakoid grana preparations from dark-adapted leaves of the different *A. thaliana* strains were inspected by high-resolution electron microscopy. The *Lhcb8-only* genotype displayed an unaltered PSII SC macro-organization with randomly distributed stain-excluding particles in a negatively stained background, similar to the *wild type* and *Lhcb4.1-only* line. As previously mentioned in ref. 27, the *koLhcb4 + Lhcb4.1$_{H242L}$* and *koLhcb6* mutants displayed highly ordered arrays of PSII particles, indicating a $C_2S_2$ head-to-tail organization. Twenty-five micrographs were acquired from 3 independent plants per each genotype. The experiment was repeated 2 times with similar results.

restored the NPQ amplitude to a lower value with respect to *Lhcb4.1-only* or the *WT* (Fig. 6a). When the Lhcb8 protein accumulation levels of individual plants from the T2 *Lhcb8-only* segregating population were plotted against their NPQ activity, we observed an intermediate trend compared to the values of the *Lhcb4.1-only* and *koLhcb4+Lhcb4.1$_{H242L}$* genotypes (Fig. 6b). In contrast, the *Lhcb4.1-only* plants achieved full NPQ recovery. We tested whether reduced NPQ could be caused by lower PsbS levels[29] by measuring PsbS accumulation immunologically (Fig. 6c) and found *WT* levels of this protein in two independent *Lhcb8-only* lines, implying that reduced NPQ in *Lhcb8-only* plants was unrelated to PsbS accumulation. An alternative explanation for the reduced NPQ could be a reduced acidification capacity of the thylakoid lumen upon excess light exposure and lower violaxanthin de-epoxidase activity. Both hypotheses were tested by measuring the de-

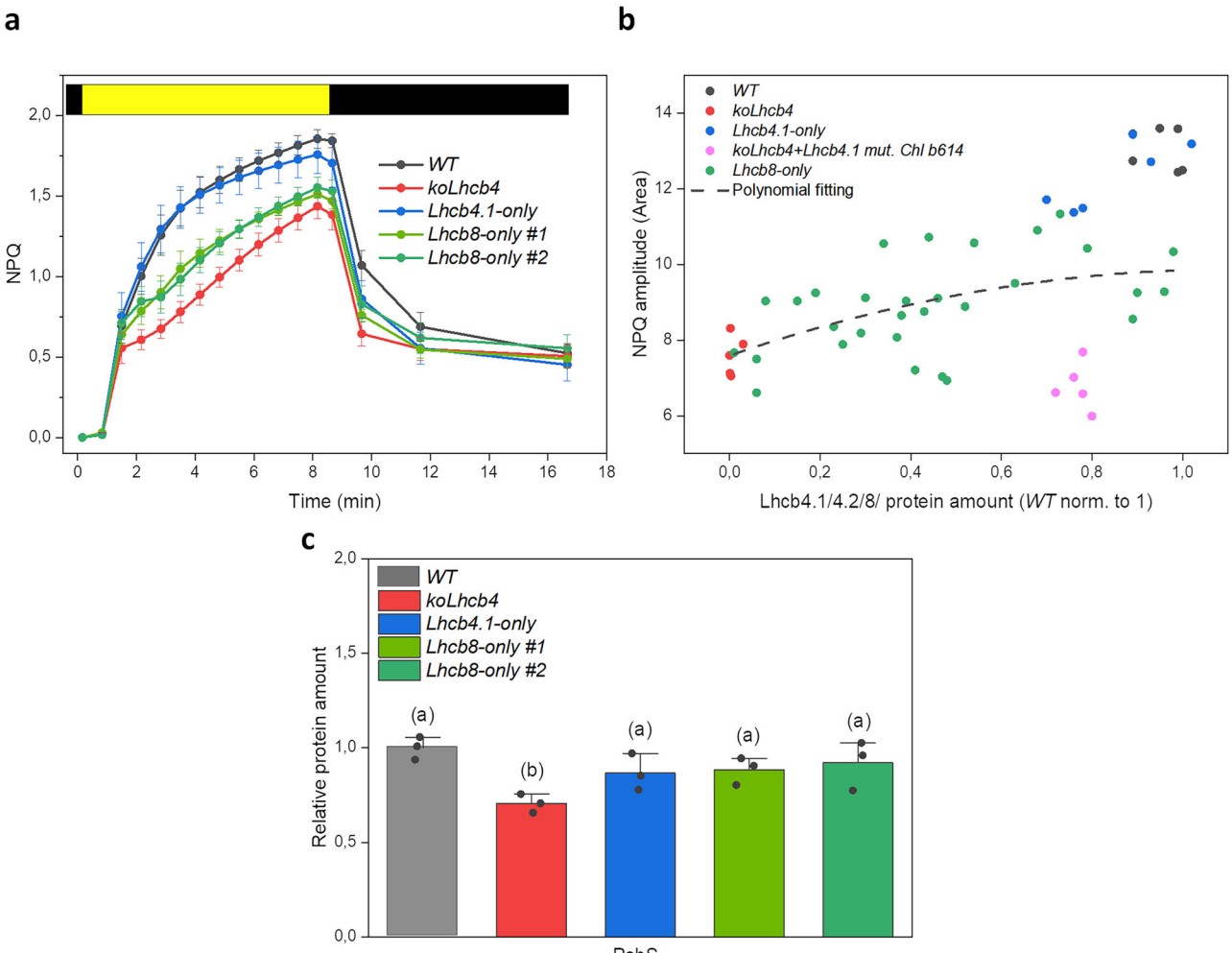

**Fig. 6 | Non-photochemical quenching (NPQ) amplitude and endogenous PsbS levels. a** Induction and relaxation kinetics of NPQ measured in the *WT*, *koLhcb4*, *Lhcb4.1-only,* and *Lhcb8-only* lines under actinic light (1280 µmol of photons m$^{-2}$ s$^{-1}$) followed by recovery in the dark (biological replicates ≥ 5). **b** Correlation between protein amount and the NPQ parameter (recorded during the light phase and calculated as the integral between minute 0 and minute 8 of the NPQ function). The levels of Lhcb4, Lhcb4$_{H242L}$, and Lhcb8 were estimated in T2 segregating populations via immunodecoration with specific antibodies and plotted against the recorded NPQ value. A quadratic function described the relationship. The *WT* Lhcb4 protein level was normalized to 1. **c** Quantification of PsbS levels by densitometric analysis of immunoblots incubated with α-PsbS antibody. Data are shown as mean ± standard deviation of $n = 3$ biological replicates. The statistical significance was determined by a one-way ANOVA test followed by the Tukey's test and depicted with lower-case letters (*P*-value ≤ 0.05).

epoxidation profile of violaxanthin (Supplementary Fig. S13). Nevertheless, the violaxanthin de-epoxidation kinetics upon exposure to high light did not show any significant alteration among genotypes.

## A structure-function analysis revealed a unique pigment organization in Lhcb8

The cryo-EM structure of the C$_2$S$_2$ supercomplex (Fig. 7a) revealed that Lhcb8 contained 12 Chls (9 Chl *a* and 3 Chl *b*) and three carotenoids, namely 1 lutein in the L1 site, 1 violaxanthin in the L2 site and 1 neoxanthin in N1 site[30]. Lhcb4.1, on the other hand, contained all the pigments found in Lhcb8 and two additional Chls (1 Chl *a* in the 613 site and 1 Chl *b* in the 614 site) in the C-terminal region (Fig. 7b, c). The pigment content of Lhcb8 and Lhcb4.1 in the cryo-EM structures was consistent with those measured by HPLC (Supplementary Table S1). Chl *a*616, located at the interface between Lhcb8 and CP47, might be partially lost in the isolated Lhcb8 complex during purification. As a result, the purified Lhcb4.1 and Lhcb8 samples contained slightly less Chls (Supplementary Table S1, based on HPLC analysis) than those observed in the cryo-EM structures.

In the carboxy-terminal region of Lhcb8, the axial ligand for Chl *b*614 in Lhcb4.1 (His274) was substituted by phenylalanine (Phe274), so

that it could not bind a Chl molecule (Fig. 7d). While the axial ligand for Chl *a*613 was conserved in both Lhcb8 (Gln259) and Lhcb4.1 (Gln259), the substitution of Gly255$_{Lhcb4.1}$ and Asn270$_{Lhcb4.1}$ by Ile255$_{Lhcb8}$ and Phe270$_{Lhcb8}$ introduced bulky side chains nearby the binding site of Chl *a*613, preventing the Chl from approaching and binding to Gln259$_{Lhcb8}$ due to steric hindrance (Fig. 7d). In Lhcb4.1, Chl *a*613, *b*614 and Trp271 encircled the luminal end ring of Lut620 at the L1 site and formed van der Waals interactions with it (Fig. 7e). Such close interactions induced an evident clockwise twist of the polyene chain in Lut620 of Lhcb4.1. As both Chl *a*613 and *b*614 were absent and Trp271 was substituted by Leu271 in Lhcb8, Lut620 in Lhcb8 had a more relaxed configuration than in Lhcb4.1. Moreover, the polyene chain of Lut620$_{Lhcb8}$ formed closer π-π interactions with Chl *a*612 than the one in Lhcb4.1 (Fig. 7f). In Lhcb8, Chl *a*612 moved closer to Lut620 by 0.7 Å with respect to the one in Lhcb4.1 so that the closest distances between the two conjugated π systems of Lut620 and Chl *a*612 became smaller in Lhcb8 (3.7 *vs* 4.1 Å for the pair in Lhcb4.1). In comparison, the violaxanthin molecules (Vio621) at the L2 sites of Lhcb8 and Lhcb4.1 were also slightly different regarding the local environments and configurations. As shown in Supplementary Fig. S14a, Vio621 also has a more relaxed configuration in Lhcb8 vs Lhcb4.1, as Ala150, responsible for

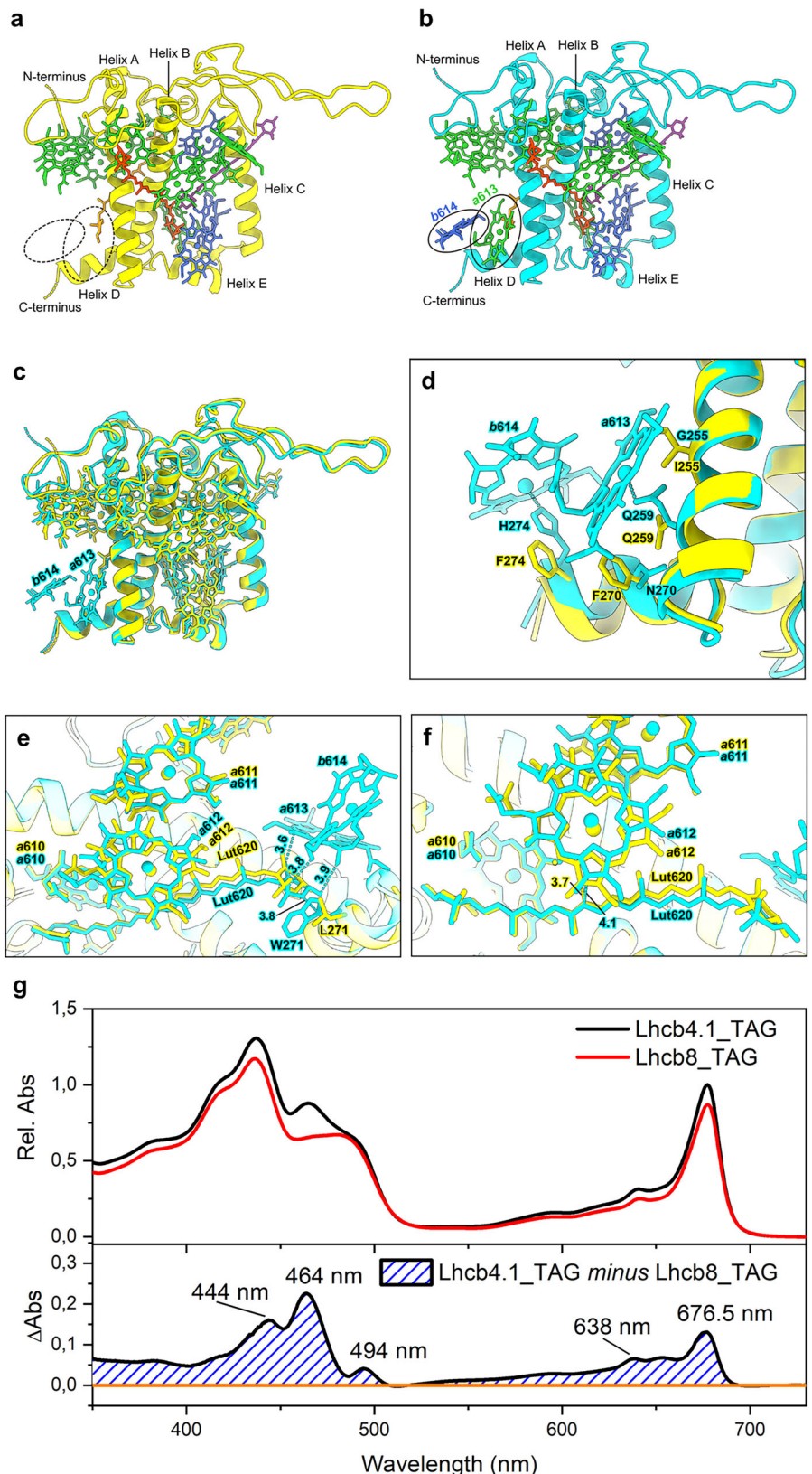

bending Vio621 at the luminal side in Lhcb4.1, was replaced by Gly151 in Lhcb8. The N1 sites in Lhcb8 and Lhcb4.1 were overall conserved, while there might be slight variations in the positions of neoxanthin and its surrounding Chls (Chl *a*604 and *b*606) due to the high mobility of the peripheral region (Supplementary Fig. S14b). Therefore, Lut620 at the

L1 site and Vio621 at the L2 site of Lhcb8 exhibited evident changes in their structure and environment relative to counterparts in Lhcb4.1.

The spectroscopical properties of the purified proteins were analysed to assess the consequences of the structural changes observed (Fig. 7g). The Lhcb4.1_TAG *minus* Lhcb8_TAG difference

**Fig. 7 | Structure, pigment composition, and linear absorption of Lhcb8 holoprotein compared with those of Lhcb4.1. a, b** The overall structure of Lhcb8 (**a**) and Lhcb4.1 (**b**) in the $C_2S_2$ supercomplexes. The apoproteins are shown as cartoon models, while the pigments are presented as stick models. Color codes: Lime green, Chl *a*; Royal blue, Chl *b*; Orange-red, Violaxanthin (Vio); Orange, Lutein (Lut); Purple, Neoxanthin (Neo). **c** Superposition of Lhcb8 with Lhcb4.1. Color codes: Yellow, Lhcb8; Cyan, Lhcb4.1. **d–f** The local regions of Lhcb8 and Lhcb4.1

exhibiting evident pigment composition or configuration differences. The carboxy-terminal regions of Lhcb8 and Lhcb4.1 are shown in (**d**), while Lut620 at the L1 site and nearby chlorophylls in Lhcb8 and Lhcb4.1 are presented in (**e**). A zoom-in view of the π-π interactions between Lut620 and Chl *a*612 is presented in (**f**). **g** Absorption spectra of Lhcb8_TAG and Lhcb4.1_TAG holoproteins isolated from plant (upper panel). The spectra were normalised on the Chls number and were used to calculate the difference spectrum (bottom panel).

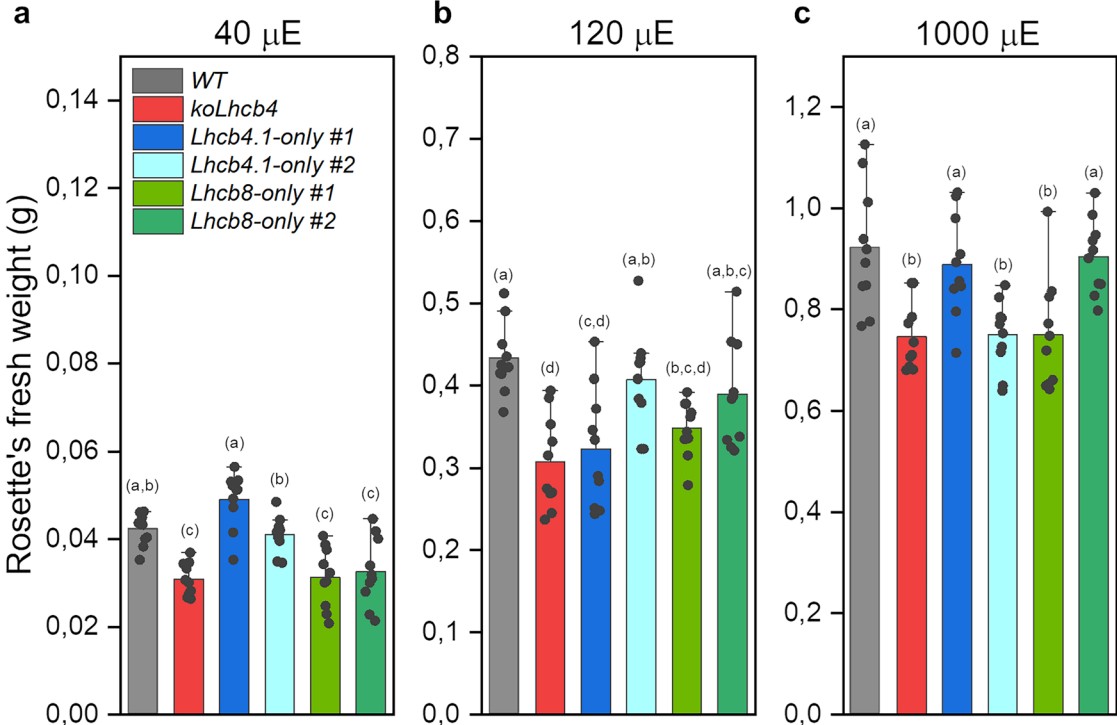

**Fig. 8 | Growth trial of selected genotypes in low light, control light, and high light conditions. a** Fresh weight of plant rosettes harvested after 2 weeks of growth under control light conditions and 4 weeks under low light conditions (40 µmol photons $m^{-2} s^{-1}$, 23 °C, 70% relative humidity, 8/16 h of day/night). **b** Fresh weight of plant rosettes harvested after 6 weeks of growth under control light conditions (120 µmol photons $m^{-2} s^{-1}$, 23 °C, 70% relative humidity, 8/16 h of day/night). **c** Fresh weight of plant rosettes harvested after 2 weeks of growth under control light

conditions and 4 weeks under high light conditions (1000 µmol photons $m^{-2} s^{-1}$, 23 °C, 70% relative humidity, 8/16 h of day/night). Data are shown as mean ± standard deviation of $n = 10$ biological replicates. The statistical significance was determined by a one-way ANOVA test followed by the Tukey's test and depicted with lower-case letters (*P*-value ≤ 0.05). The experiment was repeated 2 times with similar results.

spectrum highlighted peaks in the Q and Soret bands corresponding to Chl *a* and Chl *b* absorption (Fig. 7g). An additional positive signal was observed at 494 nm, tentatively attributed to a different environment of lutein in site L1 of Lhcb8_TAG compared to the Lhcb4.1_TAG isoform[31]. The different pigment organization of Lhcb8_TAG did not result in detectable differences in the steady-state fluorescence emission spectrum at room temperature nor at 77 K (Supplementary Fig. S15a, b), implying the terminal emitter was the same in the two complexes.

### Growth of *Lhcb8-only* plants under different conditions

We analysed the growth phenotypes of two independent *Lhcb8-only* T3 homozygous lines. Given the light intensity-dependent upregulation of the *Lhcb8* gene reported previously[16,32], we assessed plant growth under low light (40 µE, LL), control light (120 µE, CL), and high light (1000 µE, HL) conditions. As shown in Fig. 8a–c, the *koLhcb4* line exhibited significantly reduced biomass accumulation compared to the *WT* across all light conditions (LL, CL, and HL). In the *Lhcb8-only* lines, biomass production was reduced by approximately 30% relative to both the *WT* and *Lhcb4.1-only* lines under LL conditions only (Fig. 8a) and approached the *WT* level with increasing light intensity. Also,

growth under Intermittent light conditions (IL) and in the low temperature[33] showed no significant variations compared to the *WT* (Supplementary Fig. S16).

### The evolution of Lhcb4-like protein: from Chlorophytes to higher plants

The *Lhcb4* gene sequence is prevalent across all eukaryotic photosynthetic organisms[34] and gene duplication events have expanded the number of Lhcb4-encoding genes[35]. In the unicellular alga *Chlamydomonas reinhardtii*, a single gene encodes Lhcb4[36]. In the moss *Physcomitrium patens*, there are two genes[37], while in *A. thaliana*, as previously mentioned, Lhcb4 proteins are encoded by three distinct genes[38]. The phylogenetic reconstruction of Lhcb4 proteins indicated that Lhcb4.1 and Lhcb4.2 are evolutionarily closer to the algal Lhcb4 (branch length 1.31 and 1.14, respectively), suggesting that Lhcb8 could have evolved more recently (branch length 1.38) (Fig. 9a, d). However, the multiple sequence alignment of mature Lhcb4 proteins from *Arabidopsis* (*At*Lhcb4.1, *At*Lhcb4.2, *At*Lhcb8) and *Chlamydomonas* (*Cr*Lhcb4) revealed the presence of 14 conserved residue identities/deletions uniquely conserved between *At*Lhcb8 and *Cr*Lhcb4, at variance with *At*Lhcb4.1/4.2 (Fig. 9b) indicating a possible direct

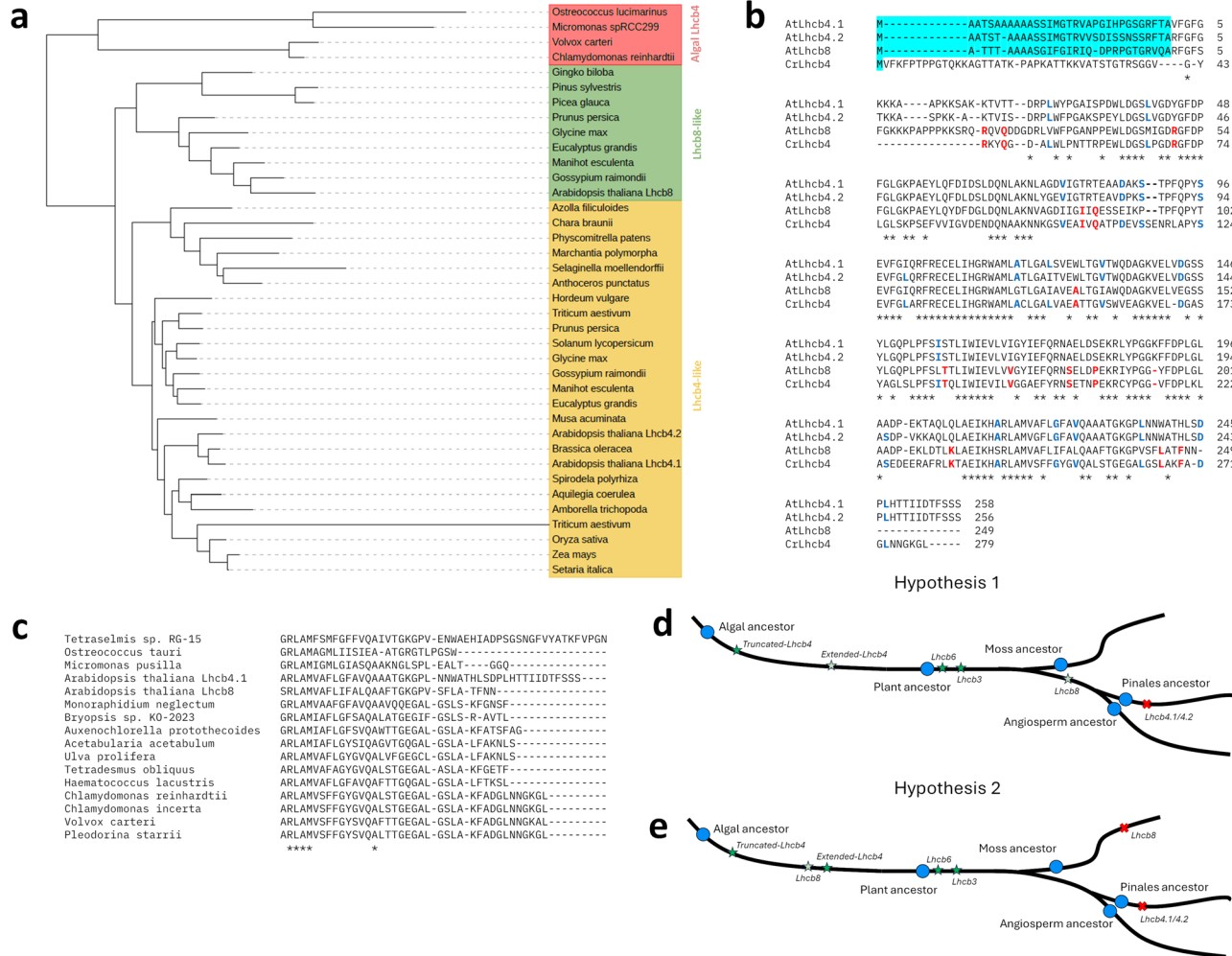

**Fig. 9 | Phylogenetic reconstruction and multiple sequence alignment of Lhcb4 proteins. a** Phylogenetic reconstruction of Lhcb4 proteins. The algal Lhcb4 proteins sequences are labelled in red, the Lhcb8-like sequences are in green, and the Lhcb4-like sequences are in yellow. **b** Multiple amino acid sequence alignment of *At*Lhcb4.1, *At*Lhcb4.2, *At*Lhcb8, and *Cr*Lhcb4 polypeptides using the T-Coffee algorithm. The 14 residues or deletions uniquely conserved between *At*Lhcb8 and *Cr*Lhcb4 are highlighted in red and the residues uniquely conserved among *At*Lhcb4.1, *At*Lhcb4.2, and *Cr*Lhcb4 are in blue. The cleaved N-terminus is highlighted in cyan. **c** Multiple sequence alignment between *At*Lhcb4.1, *At*Lhcb8 and

Lhcb4-*like* protein sequences from 14 chlorophytes (only the C-terminal region is shown). **d** Schematic representation of the evolutionary trajectory of Lhcb4, Lhcb8, Lhcb6, and Lhcb3 in which Lhcb8 are more recently evolved or (**e**) in which Lhcb8 is the ancestral isoform of Lhcb4 protein family. Blue dots represent the ancestral species; green stars denote the appearance of new gene sequences; light green indicates the sequence modifications due to natural mutations; and red crosses signify gene loss. The black line in (**d** and **e**) is not indicative of the temporal scale.

descendant (Fig. 9e). Thus, an ancestral origin of Lhcb8 cannot be fully excluded. Also, most Lhcb4 sequences from green algae possess a truncated C-terminal region similar to that in *At*Lhcb8 (Fig. 9c), indicating a strong functional correlation among these isoforms. Notably, the longer C-terminus correlates with the presence of Lhcb6 in plants and, indeed, Lhcb6 is absent in algae.

## Discussion

The LHC protein family descends from a common ancestor and shares similar functions and polypeptide sequences[39]. LHC proteins bind chlorophylls and xanthophylls and are involved in light harvesting and photoprotection through the formation of monomeric, dimeric, and multimeric complexes[40]. Monomeric LHCs (Lhcb4/5/6) are found within PSII supercomplexes, where they connect the PSII core to peripheral LHCII trimers, each establishing specific interactions with partners' subunits[41]. In this work, we aim to understand the reasons for this multiplicity of genes encoding Lhcb4 and the physiological consequences of their alternative expression.

### The C-terminal region of Lhcb8 is a structural determinant in reshaping PSII SC

Structural elucidation of the $C_2S_2M_2$ PSII SC of *Pisum sativum* revealed that at least two contact points between Lhcb4 and Lhcb6 mediates the Lhcb6 binding to the PSII SC[41]. The first is located at the C-terminal region on the stromal side, and the second at the N-terminal region on the lumenal side. The amino- and carboxy-terminal regions (as well as Chl *a*601 and Chl *b*614 associated with them) of Lhcb4 form close interactions with the AC loop-helix C (stromal half) and helix C (lumenal half)−BC loop regions of Lhcb6, respectively. Notably, Chl *b*614 from Lhcb4 is in van der Waals contacts with both the luminal half of helix C from Lhcb6 and Chl *a*614 from Lhcb3 of M-LHCII[41]. Experimental evidence in *A. thaliana* mutant Lhcb4$_{H242L}$ lacking the Chl *b*614 suggested the pivotal role of this Chl, located in the C-terminal tail of Lhcb4 (lumenal side), in mediating Lhcb4-Lhcb6 interaction[27]. The absence of Lhcb6 protein in *Lhcb8-only* lines can be attributed to the absence of the Chl *b*614 in Lhcb8. However, other stabilizing contact points located in the C-terminus tail of Lhcb4 but absent in Lhcb8

could further be responsible for shaping the Lhcb6 docking site. Similarly, the involvement of Lhcb6, together with Lhcb4, in mediating the stable connection of M-trimer to the PSII SC via the establishment of interactions of Lhcb6 and the Lhcb3 subunit of M-trimer was previously reported[41]. The reduced level, yet not the total deletion, of Lhcb3 in the *Lhcb8-only* mutant suggests that the M-trimer, containing Lhcb3, can accumulate in the thylakoid membrane without a stable association to the PSII SC, in accordance with its dissociation in high-light conditions[42].

It is interesting to note that Lhcb8 is only expressed in excess light, a condition that causes a decrease in the PSII antenna size, including loss of Lhcb6[43]. It thus appears that the replacement of Lhcb4.1/4.2 by Lhcb8 is part of the high-light acclimation process, reducing the photon harvesting capacity of plants to prevent photo-oxidation.

### The physiologic effects of Lhcb8-induced PSII reshaping

The quantum yield of photosystem II ($F_v/F_m$) was rather similar in *Lhcb8-only*, *WT,* and *Lhcb4.1-only* plants, indicating that the reshaped PSII, Lhcb8-induced, has similar photochemical activity compared to the Lhcb4.1-containing PSII SC. Instead, *Lhcb8-only* plants showed a reduced NPQ activity, despite both PsbS levels and violaxanthin-to-zeaxanthin conversion rates were unaffected. This effect could be due to the changes in the structure of Lhcb8 with respect to Lhcb4.1, which could not undergo a conformational transition to its quenching state. The site of quenching in Lhcb4 was previously identified[22] to involve a pigment cluster including Chl $a603$, Chl $a609$ and violaxanthin/zeaxanthin in site L2. Based on previous work[44] the distance/orientation between pigment in this cluster are critical for the formation of a charge-transfer state responsible for Chl to xanthophyll energy transfer. The different conformation of the xanthophyll in site L2 in Lhcb4 vs Lhcb8 might be responsible for the decreased amplitude of NPQ in *Lhcb8-only* lines.

Alternatively, this might be due to the size reduction of one of the two quenching clusters produced by the PsbS-dependent dissociation of the Lhcb4-Lhcb6-LHCII pentameric complex upon high-light exposure[42,45]. This latter hypothesis is supported by the shape of the NPQ rise curve, which closely features that of *koLhcb4*[45], with a slow rise in the early phase of quenching[46]. A decreased NPQ activity might appear inconsistent with the need for photoprotection under high light. However, NPQ is especially suited for photoprotection under rapidly changing light, while photoacclimation requires a longer time span. Our results underline the independence of these two mechanisms and the involvement of Lhcb8 in the photoacclimation process. Notably, Lut620 and Vio621 in Lhcb8 have more relaxed configurations with respect to the corresponding ligands in Lhcb4.1 (Fig. 7e, f). The different configurations of Lut620, induced by the distinct local environments (Fig. 7f), were most likely responsible for the appearance of the 494 nm feature in the difference spectrum shown in Fig. 7g. One can speculate that this modified xanthophyll configuration in Lhcb8 might enhance the coupling between the conjugated π-systems of Lut620/Vio621 and the nearby chlorophyll molecules and dissipate excess energy effectively within the antenna complex under excess light. Further studies are needed to clarify this hypothesis.

One additional regulation of the light harvesting function has been proposed based on the formation of regular arrays of PSII SC within the thylakoid membrane under low light conditions[47–49]. Incorporation of PSII SC into the semi-crystalline array may significantly enhance the light-harvesting cross-section of each PSII, enabling more efficient photon energy capture under low light conditions[50]. Array formation, however, entrains a limitation in PQ diffusion, thus restricting photosynthetic electron flow, particularly with increasing light intensity[27,28]. Since Lhcb8 fully prevented array formation in the absence of Lhcb6 (Fig. 5f, g), we suggest that the enhanced strength of the interaction between CP47 and Lhcb8, compared to the CP47-

Lhcb4.1, is effective in changing the orientation of the subunits with respect to the CP47, preventing the formation of contact points between adjacent PSII $C_2S_2$ SCs and the formation of PSII arrays. This, in turn, prevents the limitation in the accessibility of PQ at the PSII acceptor side. Moreover, previous work has shown that two adjacent $C_2S_2$ supercomplexes in the array were assembled through close contact between their Lhcb4s, with the carboxy-terminal region of Lhcb4 likely located at their interface[51]. The dramatic changes in the carboxy-terminal region of Lhcb8 (relative to Lhcb4.1) may be unfavourable for the assembly of $C_2S_2$ into arrays.

One factor triggering array formation is supposed to be the absence/reduction of Lhcb6 (Fig. 5c, e)[27,28,52]. The latter is also one of the most striking effects of acclimation to HL[43]. However, this is expected to entrain array formation, thus inducing acceptor side limitation and photoinhibition[20]. We propose that the expression of Lhcb8 in HL, to replace Lhcb4.1 and Lhcb4.2, prevents the formation of arrays in HL, thus avoiding limitations of photochemistry.

The structural elucidation of Lhcb8-$C_2S_2$ SC suggests that the peripheral antenna complexes connect with the PSII core more loosely than in the Lhcb4.1-$C_2S_2$ (Fig. 3b). It is worth noting that the smaller antenna size and the looser arrangement of the peripheral antenna of the Lhcb8-SCs vs Lhcb4.1-SCs are consistent with the slower growth rate of *Lhcb8-only* compared to *Lhcb4.1-only* in limiting light conditions but not in CL and HL conditions (Fig. 8).

### The *Lhcb8-only* PSII supercomplex is gymnosperm-like

Strikingly, the high light-induced expression of Lhcb8 in *Pisum sativum* also resulted in a strong reduction of both Lhcb6 and Lhcb3 proteins[19], suggesting that the *Lhcb8-only* genotype constitutively displayed a PSII SC configuration similar to the one acclimated to excess light[19]. The *Lhcb8-only* genotype displayed a $C_2S_2$ organization of PSII under control light. This static arrangement is typical of the gymnosperm *Picea abies*[25], which underwent a secondary loss of *Lhcb3, Lhcb4.1, Lhcb4.2,* and *Lhcb6* genes[34,53]. Also, the Lhcb8-$C_2S_2$ from *P. abies*[25] shared a loose antenna architecture with the Lhcb8-$C_2S_2$ from *A. thaliana* (Supplementary Fig. S17). This loose and expanded architecture appeared to be a specific feature of Lhcb8-$C_2S_2$ due to the adaptation of peripheral antenna domains induced by the association of Lhcb8 with CP47. The observation implies that the $C_2S_2$ SC may rearrange its antenna domains according to the specific isoform changes at the Lhcb4-binding site. However, in contrast to *Picea abies* $C_2S_2$, which was found to possess S-homotrimers composed by Lhcb1 only[25], our structural analysis revealed that in *A. thaliana*, the monomeric subunit of S-LHCII interacting with Lhcb8 and CP43 is Lhcb2, whereas the one associated with Lhcb5 is Lhcb1 (Fig. 3a).

The phylogenetic analysis suggests that Lhcb8 is evolutionarily more recent. However, some residues/deletion uniquely shared between *Cr*Lhcb4 and *At*Lhcb8 suggest that Lhcb8 may represent the ancestral gene isoform from which Lhcb4.1 and Lhcb4.2 evolved more recently (Fig. 9). In higher plants, the evolution of Lhcb4.1, with its extended C-terminal tail enabled the formation of a docking site for Lhcb6 and creating the condition for Lhcb6 appearance. This, in turn, facilitated the emergence of its docking partner, Lhcb3, the most recently evolved antenna protein[53]. The evolutionary events described in Fig. 9d, e may have contributed to the current PSII conformation observed in higher plants. The presence of both Lhcb4 and Lhcb8 in angiosperms could represent an evolutionary adaptation, allowing for more flexible and dynamic photo-acclimation, which enables PSII to respond to fluctuations in light intensity by adjusting its antenna size.

### Utilizing Lhcb8 to develop a smart canopy

Few studies have explored the selective expression of *Lhc* isoforms to improve the ability of plants to tolerate abiotic stress[54]. In this work, we investigated the rarely expressed *Lhcb8* isoform since it is transcriptionally activated in response to persistent abiotic stresses in

various angiosperms[16,39,55]. Furthermore, *Lhcb8* is the only extant *Lhcb4* ortholog in the gymnosperm *Picea abies*, an evergreen species characterized by the exclusive presence of $C_2S_2$ PSII SC configuration[25] and exhibiting peculiar photoprotective responses during overwintering[56]. The specific accumulation of Lhcb8 among the Lhcb4-like proteins underscores its role in (i) reducing the size of the PSII antenna, potentially easing excitation pressure on PSII by limiting photon absorption, and (ii) preventing the formation of extensive PSII arrays within the grana membranes resulting from Lhcb6 downregulation. These functions depend on the distinctive C-terminal and N-terminal domains of Lhcb8. Our findings highlight the potential of utilizing Lhcb8 to develop a smart canopy, which could optimize light capture and energy efficiency in plants[57].

## Methods

### Plant material
*A. thaliana* genotypes used in this work include the knock-out (ko) mutant *koLhcb4* plants (*koLhcb4.1 × koLhcb4.2 × koLhcb4.3*) previously described[20]; *koLhcb4* complemented with *Lhcb4.1* reported in ref. 22; *koLhcb4* complemented with *Lhcb4.1* carrying the H242L substitution (located in the Chl *b*614 binding site) described in ref. 27; *koLhcb4* complemented with *Lhcb4.1_TAG*, detailed in ref. 58. Lines expressing Lhcb8 and Lhcb8_TAG were obtained through *Agrobacterium*-mediated transformation of the *koLhcb4* background[59].

### Molecular biology–cloning and plant transformation
The *Lhcb8* gene, including 234 bp downstream of the stop codon, was amplified from cDNA to get the intron-less sequence. cDNA was obtained from *wild type* plants that had been adapted for 5 days to high light (1000 µmol photons $m^{-2}$ $s^{-1}$) using Nuclozol. This gene was then fused via ligation-PCR to the *Lhcb4.1* promoter (1095 bp). The resulting fusion product was cloned into the plant destination vector pBI121 using SbfI (R3642, New England Biolabs) and PacI (R0547, New England Biolabs) restriction enzymes, followed by ligation with T4 DNA Ligase (EL0011, Thermo Fisher Scientific) (see Supplementary Table S2). A 4xglycine followed by a 6xHisTAG was added using the Q5 site-directed mutagenesis kit (NEB) using specific primers (see Supplementary Table S2). After *Agrobacterium*-mediated transformation, seedlings were screened on MS medium agar plates containing kanamycin (50 mg/L) as a selection marker. Independent transformants (T1 generation) were self-fertilized for each genotype, and homozygous lines were confirmed in the T3 generation.

### Growth conditions
*Wild type* and mutant genotypes were grown in soil in a phytotron for 6 weeks at 120 µmol photons $m^{-2}$ $s^{-1}$, 23 °C, 70% relative humidity, 8/16 h of day/night. All biochemical and physiological analyses were conducted on plants prior to flowering. Plant growth was assessed by measuring the fresh weight of the rosettes at the end of the growth experiments, after around 4 or 12 weeks of light or temperature treatment.

### Membranes isolation
Stacked thylakoid membranes were isolated as previously described[60].

### Gel electrophoresis and immunoblotting
SDS-PAGE analysis was conducted using the Tris-Tricine buffer system[61]. Acrylamide gels were stained with Coomassie blue R250. For immunotitration, thylakoid samples were loaded, and proteins were electroblotted onto nitrocellulose membranes. Proteins were detected using primary antibodies α-Lhcb4 made in-house (immunogen: CP29 purified from *Z. mays*). Working dilution 1:700; Lhcb1 (PSII antenna subunit) from Agrisera (AS01 004). Working dilution 1:5000; Lhcb2 (PSII antenna subunit) from Agrisera (AS01 003). Working dilution 1:3000; Lhcb3 (PSII antenna subunit) from Agrisera (AS01 002).

Working dilution 1:2000; Lhcb6 (PSII antenna subunit) made in-house. Working dilution 1:1000; CP47 (subunit of PSII core complex) from Agrisera (AS04038). Working dilution 1:4000; CP43 (subunit of PSII core complex) from Agrisera (AS11 1787). Working dilution 1:10,000; PsaA (subunit of PSI core complex) from Agrisera (AS06 172). Working dilution 1:3000; Cyt f (subunit f of the Cyt $b_6f$) made in-house. Working dilution 1:700; PsbS protein, homemade. Working dilution 1:1000; PsbP (OEC subunit) made in-house. Working dilution 1:1000. An alkaline phosphatase-conjugated secondary antibody (Sigma-Aldrich A3687) was employed for detection. The Lhcb8 content in transgenic lines was quantified using an α-Lhcb4 primary antibody made in-house (immunogen: CP29 purified from *Z. mays*). Non-denaturing Deriphat-PAGE was performed as previously described[62].

### Densitometric analysis
Signal amplitudes were quantified from SDS-PAGE or immunoblot images using the GelPro 3.2 software (Bio-Rad). The Lhcb8 signal detected through immunoblot analysis was adjusted to account for the lower reactivity of the antibody against this isoform (approximately 4 times less reactive against Lhcb8 than against Lhcb4) (See Supplementary Fig. S1).

### Pigment analysis
Pigments were extracted from isolated holoproteins using 85% acetone buffered with $Na_2CO_3$, then separated and quantified via HPLC[63].

### Spectroscopy
Absorption measurements were performed at RT using an SLM Aminco DW-2000 spectrophotometer, with samples in 10 mM Hepes (pH 7.5) and 0.05% α-DDM.

### Analysis of Chl fluorescence
Chl fluorescence parameters $F_v/F_m$, and NPQ were measured on whole leaves using a PAM 101 fluorimeter (Heinz-Walz) according to[64]. Additional measurements were performed with a DUAL-PAM-100 (Walz, GmbH), set to a resolution of 20 µs. Fluorescence induction kinetic, reflecting the functional antenna size, were recorded using a custom-built instrument, applying 7 µmol photons $m^{-2}$ $s^{-1}$ of green light on dark-adapted leaves that had been vacuum-infiltrated with a buffer containing 20 mM Hepes (pH 7.0), 150 mM sorbitol, and 50 µM DCMU[65].

### Electron microscopy and image analysis
EM on isolated grana membranes was performed using an FEI Tecnai T12 electron microscope operating at an accelerating voltage of 100 kV. Membranes were isolated from 6-week-old *wild type* and mutant plants and grown under controlled conditions. Best grana patches, stained with 2% uranyl acetate, were analyzed, and PSII core positions were identified using ImageJ software[66].

### Statistical analysis
Statistical analyses were performed in Origin, employing one-way analysis of variance (ANOVA) followed by Tukey's post hoc test at a significant level of *P*-value < 0.05 (see figure legends for details). Error bars represent standard deviation. Means that do not share letters are significantly different.

### Phylogenetic reconstruction and multiple sequence alignment
Phylogenetic reconstruction was conducted with SHOOT[67] using the mature protein sequence of *AtLhcb8* (UniProt ID: Q9S7W1), and the phylogenetic tree was created with iTOL v6[68]. The gene identifier sequences used in the phylogenetic reconstruction are listed in the Supplementary Information. Multiple sequence alignments of *AtLhcb4.1* (UniProt ID: Q07473), *AtLhcb4.2* (UniProt ID: Q9XF88), *AtLhcb8* (UniProt ID: Q9S7W1), and *CrLhcb4* (UniProt ID: Q93WD2)

mature protein sequences were conducted using the T-Coffee algorithm with default parameter settings[69].

## Data availability

The composite cryo-EM maps of the Lhcb8-$C_2S_2$ and Lhcb4.1-$C_2S_2$ supercomplexes and their corresponding atomic coordinates have been deposited in the Electron Microscopy Data Bank and the Protein Data Bank under the accession codes of EMD-63168 and 9LK5 (Lhcb8-$C_2S_2$) and EMD-63167 and 9LK4 (Lhcb4.1-$C_2S_2$), respectively. The focused map with better density of Lhcb8-CP47 and Lhcb8-LHCII-CP26 have been deposited to the Electron Microscopy Data Bank with accession code EMD-63096 and EMD-63097, respectively. The consensus map of Lhcb8-$C_2S_2$ has been deposited to the Electron Microscopy Data Bank with accession code EMD-63098. The focused map with better density of Lhcb4.1-CP47, Lhcb4.1-LHCII-CP26, and Lhcb4.1-CS have been deposited to the Electron Microscopy Data Bank with accession code EMD-63165, EMD-63163, and EMD-63164, respectively. The consensus map of Lhcb4.1-$C_2S_2$ has been deposited to the Electron Microscopy Data Bank with accession code EMD-63166. All data analyzed during this study are included in this article and its Supplementary Information. Any other related data/materials are available in the Source Data file (https://doi.org/10.6084/m9.figshare.29329883) or from the corresponding authors upon request. Source data are provided with this paper.

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

## Acknowledgements

R.B. and L.D. acknowledge financial support from the European Research Council (ERC Advanced Grant 101053983-GrInSun). Z.L. received funding support from the National Natural Science Foundation of China (31925024) and the Chinese Academy of Sciences Project for Young Scientists in Basic Research (YSBR-015). Dr. Edoardo A. Cutolo and Dr. Mehrdad Jaberi are thanked for helpful discussions and advice in the phylogenetic reconstruction of Lhcb4 sub-family. We also thank L.H. Chen, X.J. Li, B.L. Zhu, B.X. Huang-Fu, T.E. Wang and other staff members at the Center for Biological Imaging, Core Facilities for Protein Science at the Institute of Biophysics, Chinese Academy of Sciences (IBP, CAS) for their support in cryo-EM sample screening and data collection, and X.B. Liang, A.J. Li, Y.D. Wang and X.Y. Liu for their technical support in sample preparation, biochemical experiments, data storage and processing.

## Author contributions

R.C. generated genotypes and performed the biochemical and spectroscopic characterization of mutants, experimental design, conceptualization, manuscript writing, and revision. A.A. made the biochemical and spectroscopic characterization of mutants, and the manuscript revision. Q.Z. prepared samples and performed the cryo-EM data collection and processing, model building and refinement, structural analysis, manuscript writing, and revision. J.S. participated in cryo-EM data processing, model building, and refinement. Z.L. did the structural analysis, wrote and revised the manuscript, and supervised

and coordinated the team on structural investigation. L.D. and R.B. organized the project and were involved in the experimental planning, data analysis, and manuscript preparation.

## Competing interests

The authors declare no competing interests.
