## [Peer review file · Nature Communications]

A stress-induced paralog of Lhcb4 controls the photosystem II functional architecture in *Arabidopsis thaliana*

Corresponding Author: Professor Roberto Bassi

Version 0:

Reviewer comments:

Reviewer #2

(Remarks to the Author)

This manuscript describes functional and structural studies on the roles of Lhcb4.1, 4.2 and 4.3 (Lhcb8) proteins in PSII. After knocking out all three Lhcb4 genes, the growth of the resultant plant became slower. After re-introducing either Lhcb4.1 or Lhcb8 gene, the growth of the resultant plant was recovered under normal or higher light conditions, whereas under low light condition, the growth of the Lhcb8-only plant became somewhat slower. The structures of Lhcb4.1-PSII and Lhcb8-PSII were solved by cryo-EM, and it was found that the Lhcb8-PSII has a somewhat looser structure compared with that of the Lhcb4.1-only or WT structures. These results are novel, and the manuscript is well organized. I have a few comments that need to be considered by the authors.

The Lhcb8-only plant seemed to have some reduced NPQ activity, albeit with unknown reasons. However, this seemed does not affect the growth of the Lhcb8-only mutant even under high light conditions, suggesting that this reduced NPQ activity is not related with the NPQ required under high light conditions. Is this true? Please explain why with this reduced NPQ activity the plant can still grow at the same rate with the WT under high light conditions.

Page 2, line 41: "PSII possesses a larger antenna system than PSI". This is not true for some of the organisms; for example, in some diatoms, PSI possesses 24 antenna subunits, which are apparently larger than PSII. This sentence needs to be rephrased.

Page 2, lines 47-49: "...evolution on land led to the expansion of the Light Harvesting Complex (LHC) multigene family...". However, in some marine algae, the number of LHC genes may be even larger than land plants. For example, in diatoms, more than 40 fcp genes have been found. This sentence should be modified.

Fig. 1d: What does the horizontal axis represent? Which one is Lhcb4.1/4.2/8?

Page 6, lines 133-136: "The fluorescence induction kinetics measured in the presence of DCMU showed a slower rise in Lhcb8-only lines compared to the WT or Lhcb4.1-only (Fig. 2b), implying a reduced PSII functional antenna size in Lhcb8-only lines as well. ". If this is true, why the KoLhcb4 mutant showed a much fast rise in the fluorescence induction kinetics? The PSII functional antenna size in the KoLhcb4 mutant should be much smaller than the WT or Lhcb4.1-only mutant.

In Fig. 3c and d, the colors of O and N atoms of the residues F95(Lhcb8), I94(Lhcb4.1), Y30(PsbH), Y93(Lhcb8), F92(Lhcb4.1), and Q223(CP47) should be made red and blue, to ensure the interactions between amino acid residues visible.

The Lhcb8-only plants lacked Lhcb6, but the Lhcb4.1-only plants retained Lhcb6. Where is this Lhcb6 in the structure of the Lhcb4.1-only plants shown in Fig. 3b, and where is it lacking in the Lhcb8-only structure? Please indicate.

Reviewer #3

(Remarks to the Author)

Comments :

The CP29 subunit, encoded by the *Lhcb4/Lhcb8* genes, plays important roles in the assembly of PSII-LHCII supercomplexes as well as in mediating energy transfer and photoprotection. In their manuscript, Caferrri et al. investigate several *Lhcb4*-related mutants of *Arabidopsis thaliana* and elucidate the cryo-EM structures of PSII-LHCII supercomplexes containing either *Lhcb4* or its stress-induced paralog *Lhcb8*. The cryo-EM map and models are of high quality, and the functional experimental data are clear and solid. This study reveals the interesting organizational changes in PSII-LHCII induced by *Lhcb8*, offering novel insights into the dynamics of PSII acclimation to diverse light conditions. Overall, this is an impressive research that integrates structural biology with the photoprotective functional studying of CP29, and it is expected to attract significant interest in the field of photosynthesis.

Before accepting this manuscript, there are a few imperfections that need to be revised :

1 The major: the discussion section 3.1- "The C-terminal region of *Lhcb8* is a structural determinant in reshaping PSII SC" was incomplete and improper. By the C2S2M2 structure in Su et al. 2017, C-terminal region of CP29 has strong interactions with *Lhcb6*, and then connects to M-LHCII. The explanations induced by structural details are missing in this part.

2 Line 440-445 Several distance changes between pigments and many conformational changes were described in Fig. 7, the relationship with NPQ should be referred to those conformational changes found in LHCII trimmers.

3 The clarity of Fig. 7 d-f is very poor. The yellow label are unclear and bad visual senses. The distances in Fig. 7f are almost invisible.

4 The comments about Fig. 9 : Label the specific and crucial residues in Fig. 9a is necessary. The identity between *AtLhcb8* and *CrLhcb4* in Fig. 9b shows 2 values with minimal differences. It is necessary to conduct a phylogenetic analysis of *Lhcb4* using more sequences, as well as make more relevance with Fig. 9c and d.

Considering the consistency, naming of peptides in Fig. 9b-c should be uniform with those in a, and the author should describe the source or Accession ID of all peptides in methods.

Legend of Fig.9c: (c) Multiple C-terminal sequence alignment between *AtLhcb4.1*----

Fig.9d : A arrow from truncated to extended *Lhcb4* is blurry, and should introduce in legend

5 Many references are jumbled and needs to be carefully revised.

Version 1:

Reviewer comments:

Reviewer #2

(Remarks to the Author)

The revised manuscript has clarified all my concerns, and I have no further comments.

Reviewer #3

(Remarks to the Author)

The revised version report that the authors have made significant improvements. The image quality has been enhanced and the references have been modified. The present manuscript have adequately responded to my previous concerns. At this stage, I do not have any further comments or suggestions.

From: Roberto Bassi

April 28th, 2025

To: The Editor, Nature Communications,

Ref: NCOMMS-25-02134

Dear Editor,

Thanks for your Letter of Mar 18th, 2025, with referee's reports and request to submit a revised version of our manuscript titled: "A stress-induced paralog of Lhcb4 controls the photosystem II functional architecture in *Arabidopsis thaliana*". We are pleased that both referees find our manuscript novel, well organized and written. Also, we are thankful for criticisms and suggestions that helped us in preparing a better version of the manuscript. In the following, please find a point-to-point answer to all queries and a revised version of the manuscript.

Reviewer #2 (Remarks to the Author)

This manuscript describes functional and structural studies on the roles of Lhcb4.1, 4.2 and 4.3 (Lhcb8) proteins in PSII. After knocking out all three Lhcb4 genes, the growth of the resultant plant became slower. After re-introducing either Lhcb4.1 or Lhcb8 gene, the growth of the resultant plant was recovered under normal or higher light conditions, whereas under low light condition, the growth of the Lhcb8-only plant became somewhat slower. The structures of Lhcb4.1-PSII and Lhcb8-PSII were solved by cryo-EM, and it was found that the Lhcb8-PSII has a somewhat looser structure compared with that of the Lhcb4.1-only or WT structures. These results are novel, and the manuscript is well organized. I have a few comments that need to be considered by the authors.

The Lhcb8-only plant seemed to have some reduced NPQ activity, albeit with unknown reasons. However, this seemed does not affect the growth of the Lhcb8-only mutant even under high light conditions, suggesting that this reduced NPQ activity is not related with the NPQ required under high light conditions. Is this true? Please explain why with this reduced NPQ activity the plant can still grow at the same rate with the WT under high light conditions.

We thank Reviewer #2 for his/her insightful observation that the Lhcb8-only lines, grew similar to WT despite incomplete NPQ restoration. While we do not have a definitive

answer to the question, our results suggest that the reduced antenna size may mitigate the effect of NPQ down-regulation and/or that the stress conditions we applied are not severe enough to induce a differential photo-inhibitory effect. Indeed, in future and ongoing work, we plan to analyse further the stress effects of Lhcb4 and Lhcb8 plants and we have a few hypotheses to be verified by using time-resolved spectroscopy both *in vitro*, on the isolated proteins and *in vivo*. Nevertheless, this is work requiring long experimental times and check of many different growth conditions. Thus, we prefer leaving it for future work. Indeed, gymnosperm species carry only Lhcb8 and appear to be highly resistant to stress. In that case, however, additional factors such as thylakoid protein phosphorylation (Grebe et al. PNAS 2020) and spill-over (Bag et al. Nat. Commun. 2020) have been reported to be involved in stress resistance of gymnosperms which may or may not be reproduced in our Lhcb8-only plants. All that should be analysed in future work in order to verify whether this property can be transferred to crop plants.

Page 2, line 41: “PSII possesses a larger antenna system than PSI”. This is not true for some of the organisms; for example, in some diatoms, PSI possesses 24 antenna subunits, which are apparently larger than PSII. This sentence needs to be rephrased.

Thank you for pointing this out. Of course, diatoms live in water with strikingly different spectral conditions (mostly blue and not far-red). Our group described the PSI-LHCI-LHCII-Lhcb9 supercomplex from *Physcomitrella*, which is, indeed, larger than PSII, and we, therefore, must agree. We rephrased the sentence into: “*PSII possesses a large antenna system that dynamically adjusts through short and long-term acclimation mechanisms to optimize the conversion efficiency of sunlight into chemical energy*”.

Page 2, lines 47-49: “...evolution on land led to the expansion of the Light Harvesting Complex (LHC) multigene family...”. However, in some marine algae, the number of LHC genes may be even larger than land plants. For example, in diatoms, more than 40 fcp genes have been found. This sentence should be modified.

Thank you for the correction. We rephrased the sentence in “*...evolution led to the expansion of the Light Harvesting Complex (LHC) multigene family...*”

Fig. 1d: What does the horizontal axis represent? Which one is Lhcb4.1/4.2/8?

The referee is right: the wording in Fig. 1d is misleading, indeed. For this reason, we rearranged the panel (d) of Fig. 1. Thank you for pointing this out.

Page 6, lines 133-136: “The fluorescence induction kinetics measured in the presence of DCMU showed a slower rise in Lhcb8-only lines compared to the WT or Lhcb4.1-only (Fig. 2b), implying a reduced PSII functional antenna size in Lhcb8-only lines as well. “. If

this is true, why the KoLhcb4 mutant showed a much fast rise in the fluorescence induction kinetics? The PSII functional antenna size in the KoLhcb4 mutant should be much smaller than the WT or Lhcb4.1-only mutant.

The observation is logical, indeed. However, when knocking out one component of the LHC system, others undergo enhanced expression. Previously, we have shown (Dall'Osto et al., 2020) that the NoM mutant, lacking monomeric LHCs, over-expresses the LHCII component to 160% with respect to WT and this is reflected by fluorescence induction kinetics with DCMU. As shown in Supplementary Fig. S4 and previously reported by de Bianchi et al. (2011), the koLhcb4 genotype overexpresses trimeric LHCII, leading to an enhanced Chl/ PSII RC ratio. However, as illustrated in Fig. 2a, the ko genotype does not form high-molecular-weight PSII supercomplexes. This suggests that all trimers (and their associated chlorophylls) remain structurally disconnected from the PSII SC and can rapidly emit fluorescence. The same holds for the NoM mutant (Dall'Osto et al., 2020).

In Fig. 3c and d, the colours of O and N atoms of the residues F95(Lhcb8), I94(Lhcb4.1), Y30(PsbH), Y93(Lhcb8), F92(Lhcb4.1), and Q223(CP47) should be made red and blue, to ensure the interactions between amino acid residues visible.

Thank you for the suggestion. It is a great idea to colour the O and N atoms in the side chain of these residues in red and blue to distinguish them from the carbon atoms (yellow/cyan). We have updated Fig. 3c and d accordingly, as shown below (Response Fig. 1) and in the revised manuscript.

Response Figure 1. The updated Fig. 3c and d showing the major changes in local interactions between, respectively, Lhcb8 and Lhcb4.1 and CP47 and PsbH at the amino-proximal regions of Lhcb8 and Lhcb4.1 with clear differences.

The Lhcb8-only plants lacked Lhcb6, but the Lhcb4.1-only plants retained Lhcb6. Where is this Lhcb6 in the structure of the Lhcb4.1-only plants shown in Fig. 3b, and where is it lacking in the Lhcb8-only structure? Please indicate.

In this work, we chose to solve the structures of C_2S_2 complexes that are the larger PSII supercomplex assembly present in both Lhcb4.1- and Lhcb8-only plants, mainly because the Lhcb8-only plants lack the C_2S_2M or $C_2S_2M_2$ -type complexes corresponding to the two higher MW bands as shown in lane 4 of Fig. 2a. As mentioned by reviewer #2, the Lhcb8-only plants lack Lhcb6 (Fig. 1c), and it was previously found that the *Lhcb6* deficient plants contain only the C_2S_2 supercomplex and lack the $C_2S_2M_2$ complex (Kovacs, L. et al. *Plant Cell*, 2006, DOI: 10.1105/tpc.106.045641) or has greatly reduced content of C_2S_2M and $C_2S_2M_2$ complexes (De Bianchi, S. et al., *Plant Cell*, 2011, DOI: 10.1105/tpc.111.087320), consistent with our observation. Therefore, the larger $C_2S_2M_2/C_2S_2M$ complexes with Lhcb6 are either absent or at very low abundance in Lhcb8-only genotype and therefore cannot be purified and studied by Cryo-EM. Meanwhile, the C_2S_2M and $C_2S_2M_2$ complexes are present in WT *Arabidopsis* (cfr lane 3 and lane 4 in the green gel of Fig. 2a) and do contain Lhcb6. The localization of the Lhcb6 within the $C_2S_2M_2$ complex was reported by Su X. et al. (*Science*, 2017; DOI: 10.1126/science.aan0327) by the cryo-EM method and originally, by Boekema et al (*Biochemistry*, 1999; DOI: 10.1021/bi9827161). It is located at the peripheral region in a cleft between Lhcb4 (CP29) and Lhcb3 of the M-LHCII trimer.

As for the PSII-LHCII supercomplex sample purified from the Lhcb4.1-only plants, we did observe the presence of C_2SM , C_2S_2M (Supplementary Fig. S7) and $C_2S_2M_2$ (Response Figure 2a-c) supercomplexes in the 3D classes of the cryo-EM data. These complexes all contain Lhcb6 associated at a site similar to the one reported previously (Su, X. et al. *Science*, 2017).

Due to the flexibility of the M-LHCII and Lhcb6 in C_2SM , C_2S_2M and $C_2S_2M_2$, their densities in the cryo-EM maps are fairly weak and the resolution was much lower than that of C_2S_2 . Meanwhile, the association of M-LHCII and Lhcb6 may induce some minor changes to the location and structure of the adjacent Lhcb4 (shown as the response Fig. 2d below). For the Lhcb8- C_2S_2 , the Lhcb6 and M-LHCII-binding sites are vacant because Lhcb8 lacks the domain (i.e. Chl *b*614 and the carboxy-terminal tail) crucial for Lhcb4.1/4.2 to bind Lhcb6 and Lhcb3. Despite Lhcb8 and S-LHCII are located at positions similar to (but slightly changed) those of Lhcb4.1 and S-LHCII in Lhcb4.1- $C_2S_2M_2$ (Response Fig. 2e). Therefore, it is better to compare the Lhcb8-containing supercomplex with the Lhcb4.1- C_2S_2 supercomplex of the same type as presented in Figure 3. Response Figure 2 has been included in the revised manuscript as the supplementary Fig. S8 and cited in the main text (lines 167-171, p6).

Response Figure 2 The C₂SM, C₂S₂M and C₂S₂M₂ supercomplexes from the Lhcb4.1-only plant. a-c, Cryo-EM maps of C₂SM (a), C₂S₂M (b), C₂S₂M₂ (c) superposed with the corresponding cartoon models. Note that the densities of M-LHCII and Lhcb6 are relatively weaker than those of S-LHCII and C₂ regions presumably due to high mobility or lower occupancy. **d and e**, superposition of Lhcb4.1-C₂S₂M₂ with Lhcb4.1-C₂S₂ (d) and Lhcb8-C₂S₂ (e) showing the spatial relationship between the binding sites of Lhcb6 and Lhcb4.1/Lhcb8. The dashed elliptical rings in panel **e** indicate the vacant Lhcb6 and M-LHCII-binding sites in Lhcb8-C₂S₂.

Reviewer #3 (Remarks to the Author):

Comments:

The CP29 subunit, encoded by the *lhcb4*/*lhcb8* genes, plays important roles in the assembly of PSII-LHCII supercomplexes as well as in mediating energy transfer and photoprotection. In their manuscript, Caferri et al. investigate several Lhcb4-related mutants of *Arabidopsis thaliana* and elucidate the cryo-EM structures of PSII-LHCII supercomplexes containing either Lhcb4 or its stress-induced paralog Lhcb8. The cryo-EM map and models are of high quality, and the functional experimental data are clear and solid. This study reveals the interesting organizational changes in PSII-LHCII

induced by Lhcb8, offering novel insights into the dynamics of PSII acclimation to diverse light conditions. Overall, this is an impressive research that integrates structural biology with the photoprotective functional studying of CP29, and it is expected to attract significant interest in the field of photosynthesis.

We thank referee #3 for her/his appreciation of our work.

Before accepting this manuscript, there are a few imperfections that need to be revised:

1 The major: the discussion section 3.1- “The C-terminal region of Lhcb8 is a structural determinant in reshaping PSII SC” was incomplete and improper. By the C₂S₂M₂ structure in Su et al. 2017, C-terminal region of CP29 has strong interactions with Lhcb6, and then connects to M-LHCII. The explanations induced by structural details are missing in this part.

Thank you to the reviewer #3 for prompting us to elucidate further this crucial aspect of the work. The text included between line 408 and 416 has been rephrased as follow: *“Structural elucidation of the C₂S₂M₂ PSII SC of Pisum sativum revealed that at least two contact points between Lhcb4 and Lhcb6 are present in mediating the Lhcb6 binding to the PSII SC (Su et al., 2017). The first is located at the stroma side and the second at the luminal side. The amino- and carboxy-terminal regions (as well as Chl a601 and Chl b614 associated with them) of Lhcb4 form close interactions with the AC loop–helix C (stromal half) and helix C (luminal half)–BC loop regions of Lhcb6 respectively. Notably, Chl b614 from Lhcb4 is in van der Waals contacts with both the luminal half of helix C from Lhcb6 and Chl a614 from Lhcb3 of M-LHCII (Su et al., 2017). Experimental evidence in Arabidopsis thaliana mutant Lhcb4_{H242L} lacking the Chl b614 suggested the pivotal role of this Chl, located in the C-terminal tail of Lhcb4 (luminal side), in mediating Lhcb4-Lhcb6 interaction (Guardini et al., 2022). The absence of Lhcb6 protein in Lhcb8-only lines can be attributed to the absence of the Chl b614 in Lhcb8. However, other stabilizing contact points located in the C-terminus tail of Lhcb4 but absent in Lhcb8 could further be responsible for shaping the Lhcb6 docking site.*

Similarly, it was previously demonstrated the involvement of Lhcb6, together with Lhcb4, in mediating the stable connection of M-trimer to the PSII SC via the establishment of interactions of Lhcb6 and the Lhcb3 subunit of M-trimer (Su et al., 2017). The accumulation reduction but not the total deletion of Lhcb3 in the Lhcb8-only mutant suggests that, unlike the absence of Lhcb4-Lhcb6 interaction in Lhcb8-C₂S₂ where Lhcb6 cannot accumulate in a free form, the M-trimer can accumulate in the thylakoid membrane without a stable association to the PSII SC, in accordance with its dissociation in high light conditions (Betterle et al. J. Biol Chem, 2009).” We are confident that this revised version of the text is now clearer and more complete.

2 Line 440-445 Several distance changes between pigments and many conformational changes were described in Fig. 7, the relationship with NPQ should be referred to those conformational changes found in LHCII trimmers.

Thank you for this consideration. After the suggestion offered by reviewer #3, we proceeded to the NPQ discussion implementation related to the Lhcb8-only lines as follow: “... The site of quenching in CP29 (Lhcb4) was previously identified By Guardini et al. (Nat. Plants, 2020) to involve a pigment cluster including Chl a603, Chl a609 and Violaxanthin/zeaxanthin in site L2. Based on previous work (Ahn et al., Science, 2008), the distance/orientation between pigments in this cluster are critical for the formation of a charge-transfer state responsible for Chl to Xanthophyll energy transfer. The different conformation of the xanthophyll in site L2 in Lhcb4 vs Lhcb8 might be responsible for the decreased amplitude of NPQ in Lhcb8-only lines”.

3 The clarity of Fig. 7 d-f is very poor. The yellow label are unclear and bad visual senses. The distances in Fig. 7f are almost invisible.

Thank you for the comment. We have updated the labels and distances in Fig. 7d-f to improve their contrast and visibility (see response Fig. 3 below).

Response Figure 3 The revised Fig. 7c-f with updated labels and distances.

4 The comments about Fig. 9: Label the specific and crucial residues in Fig. 9a is necessary. The identity between AtLhcb8 and CrLhcb4 in Fig. 9b shows 2 values with minimal differences. It is necessary to conduct a phylogenetic analysis of Lhcb4 using more sequences, as well as make more relevance with Fig. 9c and d. Considering the consistency, naming of peptides in Fig. 9b-c should be uniform with those in a, and the author should describe the source or Accession ID of all peptides in methods.

Legend of Fig.9c: (c) Multiple C-terminal sequence alignment between AtLhcb4.1----

Fig.9d: A arrow from truncated to extended Lhcb4 is blurry, and should introduce in legend

Reconstructing the evolutionary trajectory of a protein is not always straight-forward. The extended phylogenetic reconstruction of the Lhcb8 vs Lhcb4 phylogeny requested by the reviewer was indeed performed. The results indicate that Lhcb8 could be evolutionary more distant to the algal Lhcb4 respect to plant Lhcb4.1 and 4.2. This is in agreement with the conclusions from Grebe et al. (J Exp Bot, 2019) and Bag et al. (Plants, 2021). Nevertheless, we also observed 14 conserved residue uniquely shared between the algal Lhcb4 and Lhcb8, suggesting a common lineage. The strength of 14 unique conservation is high and we need to keep open the choice between the hypothesis of Lhcb8 being the ancestral form of CP29 and the alternative hypothesis of a later divergence from Lhcb4. For this reason, we modified the Fig. 9 as requested by the reviewer #3, including the caption, results and discussion introducing two possible evolutionary trajectories: the first in which Lhcb8 evolved more recently and the second in which it is an ancestral isoform.

5 Many references are jumbled and needs to be carefully revised.

We thank reviewer #3 for indicating this oversight. References were readily updated.

We are confident that all queries have been answered. We thank again the referees for their help and suggestions. Looking forward to a confirmation that the revised version of the manuscript is now suitable for publication on Nature Communications,

Yours sincerely,

Roberto Bassi